# Glucose-derived glutamate drives neuronal terminal differentiation in vitro

Laura D'Andrea [1,7], Matteo Audano[1,7], Silvia Pedretti [1], Silvia Pelucchi[1], Ramona Stringhi [1], Gabriele Imperato [1], Giulia De Cesare[1], Clara Cambria[2], Marine H Laporte [3], Nicola Zamboni [4], Flavia Antonucci [2,5], Monica Di Luca[1], Nico Mitro [1,6,8 ✉] & Elena Marcello [1,8 ✉]

## Abstract

Neuronal maturation is the phase during which neurons acquire their final characteristics in terms of morphology, electrical activity, and metabolism. However, little is known about the metabolic pathways governing neuronal maturation. Here, we investigate the contribution of the main metabolic pathways, namely glucose, glutamine, and fatty acid oxidation, during the maturation of primary rat hippocampal neurons. Blunting glucose oxidation through the genetic and chemical inhibition of the mitochondrial pyruvate transporter reveals that this protein is critical for the production of glutamate, which is required for neuronal arborization, proper dendritic elongation, and spine formation. Glutamate supplementation in the early phase of differentiation restores morphological defects and synaptic function in mitochondrial pyruvate transporter-inhibited cells. Furthermore, the selective activation of metabotropic glutamate receptors restores the impairment of neuronal differentiation due to the reduced generation of glucose-derived glutamate and rescues synaptic local translation. Fatty acid oxidation does not impact neuronal maturation. Whereas glutamine metabolism is important for mitochondria, it is not for endogenous glutamate production. Our results provide insights into the role of glucose-derived glutamate as a key player in neuronal terminal differentiation.

**Keywords** Glutamate; Local Protein Translation in Neurons; Metabolism; Mitochondrial Pyruvate Carrier
**Subject Categories** Metabolism; Neuroscience

## Introduction

Most neurological and psychiatric disorders are associated with impairments in energy metabolism (Bonvento and Bolaños, 2021). Thus, understanding the role of the different metabolic pathways during brain development, function, and aging is central for maintaining brain health.

The brain is the most energy-demanding organ in the body, as it must accomplish neuronal functions such as generation of action potentials, neurotransmitter release and uptake, and the restoration of the ion movements generated by postsynaptic currents (Alle et al, 2009). Glucose is the main fuel substrate used by neurons. After glycolysis, pyruvate is completely oxidized in the tricarboxylic (TCA) cycle associated with oxidative phosphorylation (Oxphos) to provide energy in the form of ATP to accomplish neuronal functions. In certain conditions, such as hypoglycemia, neuronal cells can take advantage of glutamate and glutamine to produce energy (McKenna, 2007). Therefore, glutamate can be used by neurons as an excitatory neurotransmitter or as an energy source through deamination to α-ketoglutarate (αKG), formation of the inhibitory transmitter γ-aminobutyric acid, and glutathione synthesis (Agostini et al, 2016). Astrocytes provide the neurons with glutamine to produce glutamate but also lactate to generate pyruvate (Bonvento and Bolaños, 2021).

Central to brain metabolism are the mitochondria. In addition to their role as the powerhouses of cells, these dynamic organelles are involved in calcium homeostasis, apoptosis, and reactive oxygen species (ROS) production and are of utmost importance in neurotransmitter synthesis and inactivation (Flippo and Strack, 2017).

Mitochondria and energy metabolism are key in the series of events leading to neuronal differentiation, including growth of the axon, dendritic arborization, and formation of functional synapses (Pekkurnaz and Wang, 2022). During the differentiation process, mitochondrial function and biogenesis are boosted to favor the shift from glycolysis to Oxphos. These modifications provide neurons with a substantial amount of energy to cope with the newly established homeostasis and morphology acquired during differentiation (Agostini et al, 2016). The interaction between mitochondria and metabolism extends beyond neurons. In a recent study, Morant-Ferrando B. and colleagues discovered that astrocytes utilize the physiological oxidation of fatty acids to induce an energetically inefficient state in the assembly of mitochondrial respiratory chain super complexes. This state promotes the

[1]Department of Pharmacological and Biomolecular Sciences "Rodolfo Paoletti", Via Giuseppe Balzaretti 9, 20133 Milan, Italy. [2]Department of Medical Biotechnology and Translational Medicine (BIOMETRA), Via F.lli Cervi 93, Segrate, 20054 Milan and via Vanvitelli 32, Milan, Italy. [3]Department of Molecular and Cellular Biology, University of Geneva, Geneva, Switzerland. [4]Institute of Molecular Systems Biology, ETH Zurich, Zurich, Switzerland. [5]Institute of Neuroscience, IN-CNR, Milan, Italy. [6]Department of Experimental Oncology, IEO, European Institute of Oncology IRCCS, Milan, Italy. [7]These authors contributed equally: Laura D'Andrea, Matteo Audano. [8]These authors contributed equally as senior authors: Nico Mitro, Elena Marcello. ✉E-mail: nico.mitro@unimi.it; elena.marcello@unimi.it

generation of essential mitochondrial ROS, crucial for sustaining cognitive performance (Morant-Ferrando et al, 2023). Despite several studies have been conducted to disentangle the role of mitochondria and metabolism in neurogenesis (Iwata et al, 2020; Petrelli et al, 2023; Iwata et al, 2023; Iwata and Vanderhaeghen, 2021), little is known about the role of the main metabolic pathways during the physiological process of terminal differentiation of hippocampal neurons (Agostini et al, 2016).

To address the metabolic changes that occur during neuronal maturation and morphogenesis, we used an in vitro model of terminal differentiation of hippocampal neurons that has been widely used to study neuronal cell biology and the mechanisms underlying the transformation of neuroblasts to mature neurons (Dotti et al, 1988). We evaluated the impact of the selective inhibition of glucose, fatty acid oxidation and glutamine oxidative and reductive pathways on mitochondrial function and how this

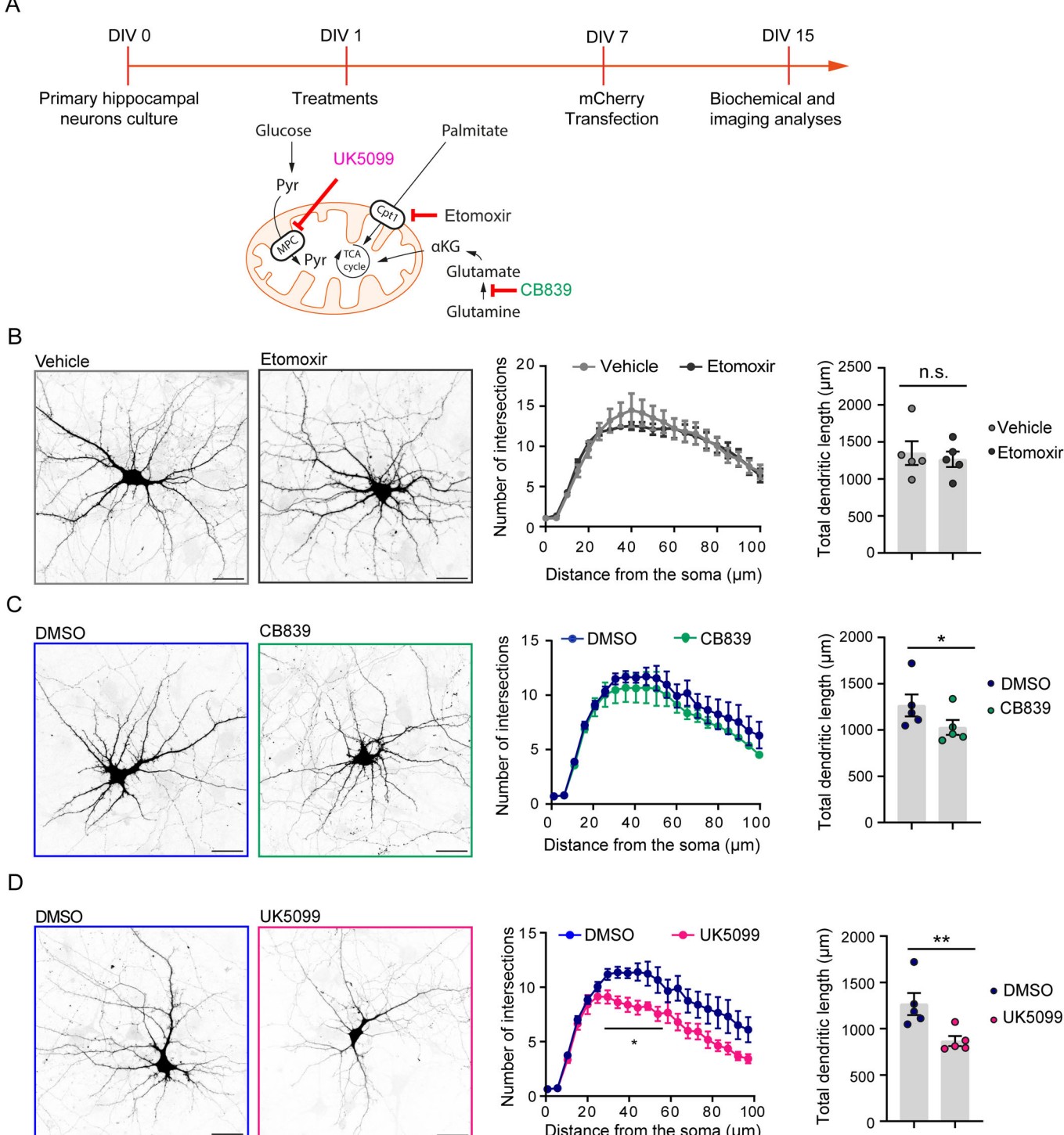

**Figure 1. The inhibition of MPC and Gls-1, but not CPT-1a, affects neuronal arborization.**

(A) Timeline of the experimental procedure. Primary hippocampal neurons were isolated and plated (DIV0) and treatments (20 µM etomoxir, 4 µM CB839, or 50 µM UK5099) were applied at DIV1. The transfection with a plasmid expressing the mCherry protein was performed at DIV7 and analysis was performed at DIV15. (B–D) Representative confocal images (left panels) of hippocampal neurons transfected with a plasmid expressing the mCherry protein and treated with etomoxir (B), CB839 (C), or UK5099 (D); water (vehicle) and DMSO were used as controls (scale bar 30 µm). For each treatment, the quantifications of the dendritic arborization (Sholl analysis) and total dendritic length are reported in the graphs on the right. (For Sholl analysis: 5 biological replicates; Vehicle: 37 neurons, etomoxir: 36 neurons, DMSO: 39 neurons, CB839: 39 neurons, UK5099: 32 neurons. Multiple t-test, degrees of freedom (df) = 8; etomoxir vs. vehicle and CB839 vs. DMSO not significant; UK5099 vs. DMSO t = 2.63 *p = 0.0301, t = 3.65 **p = 0.0064, t = 3.61 **p = 0.0068, t = 3.21 *p = 0.0124, and t = 2.57 *p = 0.0333 at 30, 35, 40, 45, and 50 µm from the soma, respectively. For dendritic length: 5 biological replicates; vehicle: 37 neurons, etomoxir: 36 neurons, DMSO: 39 neurons, CB839: 38 neurons, UK5099: 33 neurons; etomoxir vs. vehicle, paired t-test t = 0.9689, df = 4, p = 0.3875; CB839 vs. DMSO, paired t-test t = 3.582, df = 4, *p = 0.0231; and UK5099 vs. DMSO paired t-test t = 5.087, df = 4, **p = 0.0070). The control condition (DMSO) is common to CB839 and UK5099 treatments; therefore, the Sholl analysis and dendritic length data of DMSO-treated cells are reported in both panels (C) and (D). Data information: Data are reported as mean ± SE. Source data are available online for this figure.

affected neuronal arborization and synaptogenesis during terminal differentiation. Here we show that, although all the inhibitors we used affected mitochondrial function, only the inhibition of complete glucose oxidation in the TCA cycle resulted in a dramatic impairment of neuronal architecture and synaptic formation. Our data show that glucose oxidation is an obligate metabolic pathway to sustain glutamate production. The combination of a reduction in glutamate levels and altered mitochondrial function upon inhibition of glucose oxidation leads to a dramatic decrease in the energy-demanding process of protein translation. Moreover, either extracellular glutamate supplementation or the activation of group I metabotropic glutamate receptors (mGluRs) rescues synaptic transmission and restores local synaptic translation, while it has no effect on global protein synthesis. Our results provide clear evidence that endogenously synthesized glutamate derived from glucose oxidation plays a key role as a neurotransmitter required to promote differentiation.

## Results

### Distinct metabolic pathways differentially contribute to neuronal arborization and synapse formation

The development, building, and maintenance of neuronal architecture and the rapid integration of exogenous and endogenous stimuli require a high energy demand. However, the role of energy metabolism has been studied in more detail during neurogenesis than during neuronal terminal differentiation (Khacho et al, 2019; Knobloch et al, 2017; Mandel et al, 2016). Therefore, as a first step, we analyzed a published dataset to evaluate whether energy metabolism is associated with neuron differentiation (Data Ref: Hartl et al, 2008). The respiratory chain was the only metabolic pathway among the most enriched gene clusters during neuronal differentiation (Fig. EV1A, red bars).

Since the respiratory chain is fueled by the oxidation of several energy substrates, namely glucose, fatty acids, and glutamine, we explored which metabolic pathway is the main contributor to the definition of neuronal architecture and dendritic complexity, influencing neuronal maturation. For this purpose, we prepared rat primary hippocampal neurons that were differentiated for 15 days in vitro (DIV 15). At DIV1, cells were exposed to one of three different metabolic inhibitors: UK5099, an inhibitor of the mitochondrial pyruvate transporter (MPC); CB839, a glutaminase 1 (Gls-1) inhibitor; or etomoxir, which blunts the activity of carnitine palmitoyltransferase 1a (CPT-1a); DMSO (for UK5099

and CB839) and water (for etomoxir) were used as control treatments (Fig. 1A).

Considering that metabolism can have pleiotropic effects, first we assessed the impact of these chemical compounds on the proportion of cell types in our cultures. To detect the cell types in our cultures, we performed immunocytochemical analyses using specific markers: neuron-specific protein microtubule-associated protein 2 (MAP2) and Glial Fibrillary Acidic Protein (GFAP) to detect neurons and astrocytes, respectively. The statistical analysis showed that the proportion of neurons and astrocytes did not change in treated-cell cultures compared to control conditions (Appendix Fig. S1).

Next, we analyzed neuronal morphology by Sholl analysis, a common method used to quantify neuronal complexity, at DIV15 in neurons expressing the fluorescent mCherry protein as a filler to visualize neuronal structure (Fig. 1A). The results showed that etomoxir administration had no effect on neuronal arborization and dendritic length (Fig. 1B), while CB839 exposure significantly reduced dendritic length without influencing the dendritic architecture compared to control cells (Fig. 1C). Treatment with UK5099 strongly affected neuronal complexity, triggering a significant decrease in dendritic length, and reducing the complexity of dendritic arborization compared to control neurons (Fig. 1D). A milder effect on neuronal architecture was still detected when UK5099 was administered to differentiated neurons (at DIV10, Fig. EV1B–D). These observations suggest that the MPC-mediated import of pyruvate into the mitochondria plays a key role in neuronal maturation and is required for the maintenance of structural integrity in neurons. On the other hand, Gls-1 and CPT-1a inhibition did not have a macroscopic effect on the neuronal structure.

Next, we investigated the impact of metabolic inhibitors on synapse formation. We performed immunostaining for postsynaptic density 95 (PSD-95), a scaffolding element of the postsynaptic density (Coley and Gao, 2018; Won et al, 2017), in DIV15 hippocampal neurons labeled with the dendritic marker MAP2. Quantitative analysis revealed a significant reduction in PSD-95 clusters along dendrites upon UK5099 treatment (Fig. 2A). Neither etomoxir nor CB839 treatment had this effect (Fig. 2A). These results were confirmed with a biochemical approach. We evaluated PSD-95 protein levels from DIV15 neuronal cultures exposed to the metabolic inhibitors in total homogenate and in the purified Triton-insoluble fraction (TIF), which is enriched in postsynaptic proteins. PSD-95 protein levels were reduced in both total homogenate and in the TIF only upon treatment with the MPC inhibitor UK5099 (Fig. 2B,C). No changes in PSD-95 protein levels

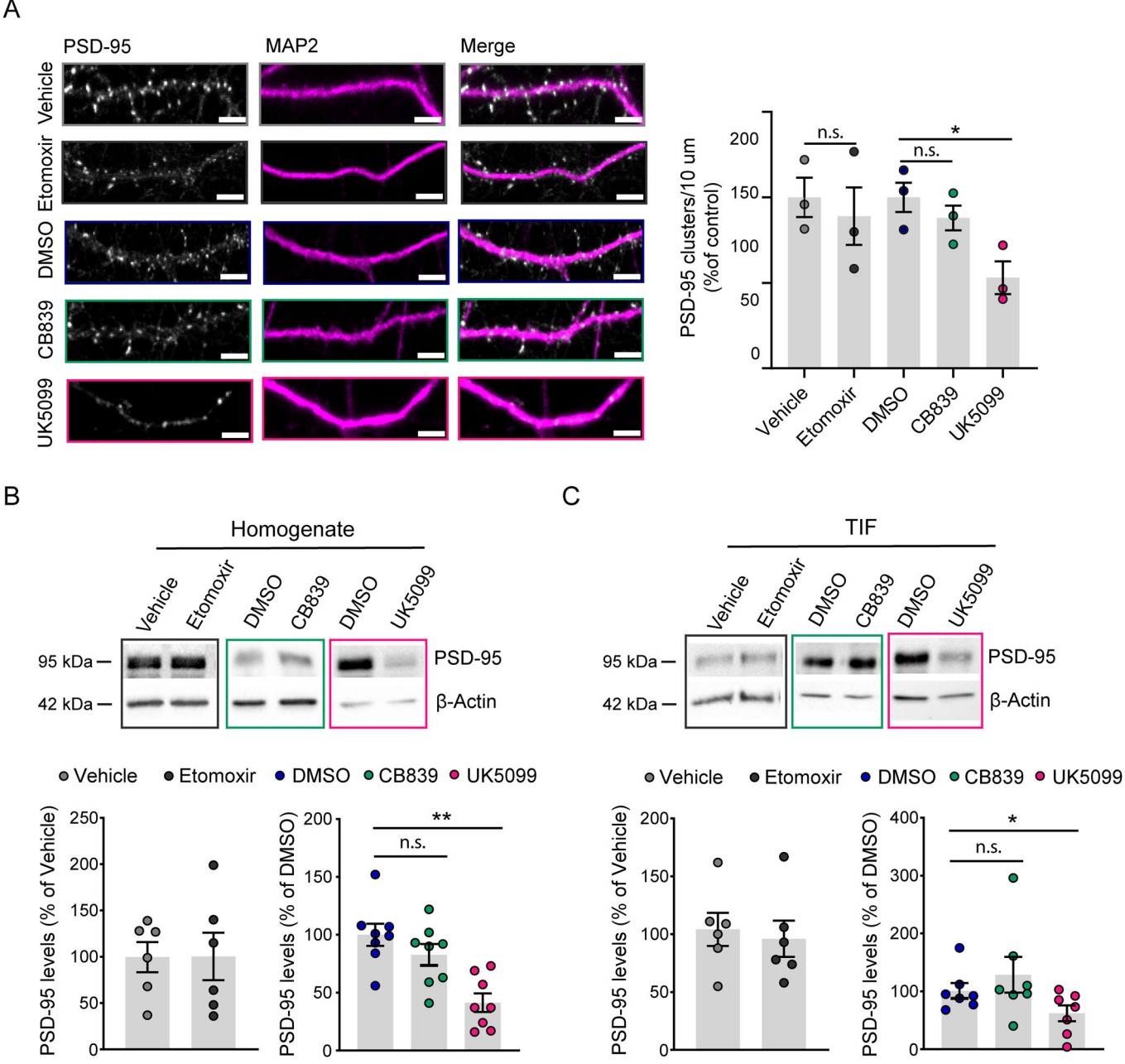

**Figure 2. Blocking the activity of MPC, but not that of CTP-1a or Gls-1, impairs synaptogenesis.**

(**A**) Representative confocal images of PSD-95 and MAP2 along dendrites of DIV15 hippocampal neurons treated with etomoxir, CB839, or UK5099; quantification of PSD-95 clusters is reported in the graph on the right (3 biological replicates; 33 neurons for vehicle 28 neurons for etomoxir; 38, 27, and 30 neurons for DMSO, CB839, and UK5099, respectively; etomoxir vs. vehicle paired t-test t = 2.079, df = 2, p = 0.1732. For CB839 vs. DMSO, paired t-test t = 1.258, df = 2, p = 0.3354, and UK5099 vs. DMSO paired t-test t = 6.225, df = 2, *p = 0.0248; scale bar 5 μm; PSD-95: gray, MAP2: magenta). (**B,C**) Representative western blot (upper panel) and quantification (lower panel) of PSD-95 levels in the homogenate (**B**) and in the Triton-insoluble fraction (**C**) isolated from neurons exposed to etomoxir, UK5099, or CB839; vehicle and DMSO were used as controls and β-Actin was used as a loading control (homogenate: n = 6 biological replicates for vehicle and etomoxir and n = 8 biological replicates for DMSO, CB839, and UK5099. Paired t-test, etomoxir vs. vehicle t = 0.03984, df = 5, p = 0.9698; CB839 vs. DMSO t = 1.322, df = 7, p = 0.2277; UK5099 vs. DMSO t = 5.272, df = 7, **p = 0.0012. Triton-insoluble fraction n = 6 biological replicates for vehicle and etomoxir, 7 biological replicates for DMSO, CB836 and UK5099; paired t-test etomoxir vs. vehicle, p = 0.7434, t = 0.3460, df = 5; CB839 vs. DMSO paired t-test, p = 0.1991, t = 1.443, df = 6; UK5099 vs. DMSO paired t-test *p = 0.0156, t = 3.341, df = 6). Data information: Data are reported as mean ± SE. Source data are available online for this figure.

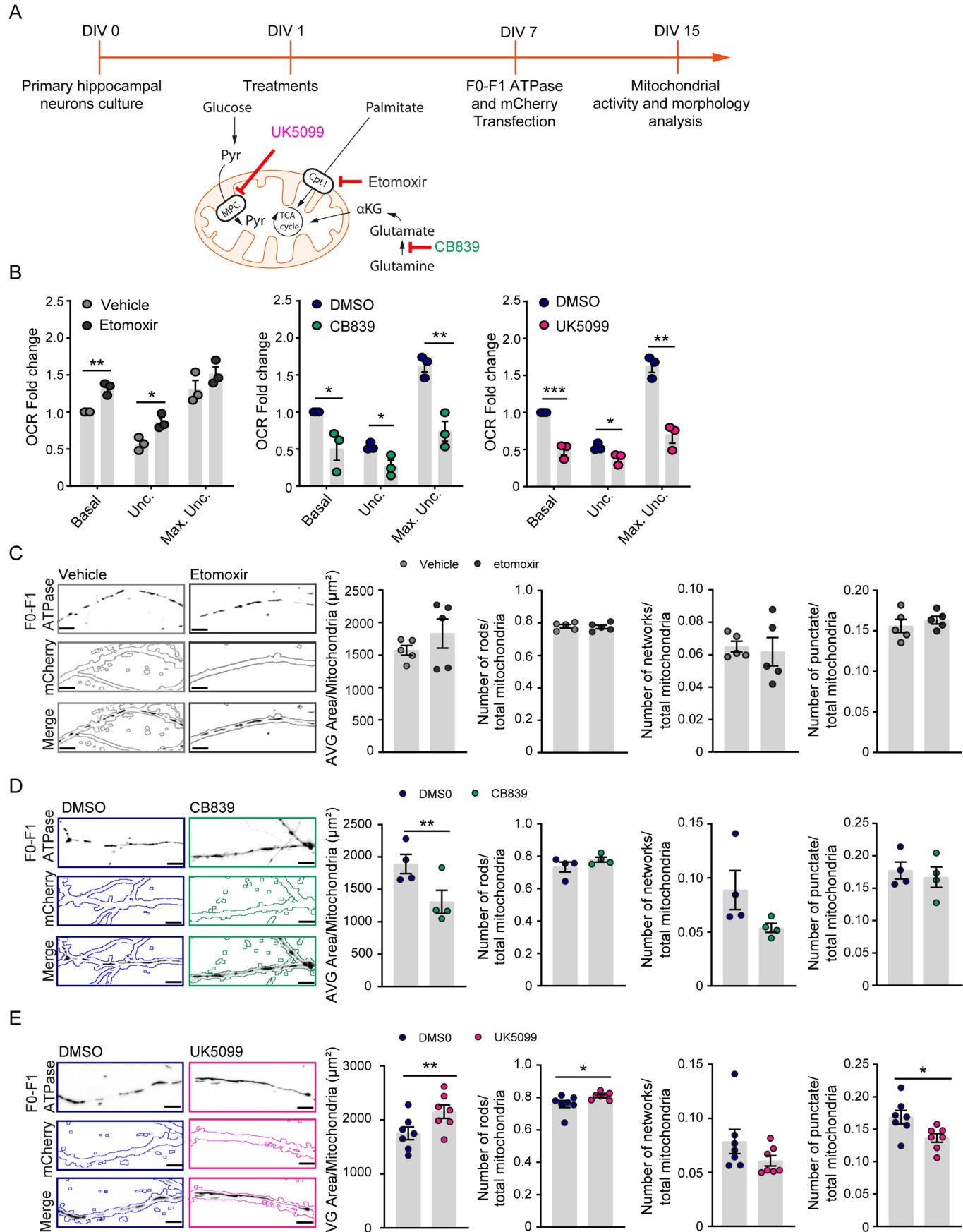

**Figure 3. Inhibition of MPC, CTP-1a, and Gls-1 function differentially affects mitochondrial activity and morphology.**

(A) Timeline of the experimental procedure. Primary hippocampal neurons were isolated and plated (DIV0) and treatments (20 µM etomoxir, 50 µM UK5099, and 4 µM CB839; vehicle and DMSO were used as controls) were applied at DIV1. The transfection with a plasmid expressing the mCherry protein together with a plasmid expressing GFP-tagged subunit 9 of the F0-F1 ATPase was performed at DIV7 and functional and morphological analyses of mitochondria were performed at DIV15. (B) Basal, uncoupled (oligomycin), and maximal uncoupled (FCCP) oxygen consumption rate analyses of DIV15 neurons treated with vehicle or the indicated metabolic inhibitors ($n = 3$ biological replicates). Data information: Data are reported as mean ± SD. The control condition (DMSO) is common to CB839 and UK5099 treatments. Statistical analysis was performed using one-way ANOVA with Dunnett's post-test. ***$p < 0.001$; **$p < 0.01$, and *$p < 0.05$ vs. vehicle or DMSO. (C–E) Representative confocal images of mitochondria in neurons transfected with mCherry (outline, light and dark gray for vehicle and etomoxir, respectively; blue for DMSO, green for CB839, and pink for UK5099) and SV9 F0-F1 ATPase (black) plasmids and treated with etomoxir (C), CB839 (D), or UK5099 (E). DMSO was used as control for UK5099 and CB839, while water (vehicle) was used for Etomoxir (scale bar 5 µm). For each condition, the graphs (from left to right) show the quantification of the average area of mitochondria in µm² and the number of rod, networks, and punctate mitochondria with respect to the total number of mitochondria ($n = 5$ biological replicates etomoxir vs vehicle, $n = 4$ CB839 vs DMSO; $n = 7$ UK5099 vs DMSO; ≥30 neurons were analyzed for each condition. Etomoxir vs. vehicle: paired t-test; average area $t = 1.240$, $df = 4$, $p = 0.2829$; rods $t = 0.3249$, $df = 4$, $p = 0.7615$; networks $t = 0.4078$, $df = 4$, $p = 0.7043$; and punctate mitochondria $t = 0.1244$, $df = 4$, $p = 0.2814$. CB893 vs. DMSO: paired t-test; average area, $t = 7.711$, $df = 3$ **$p = 0.0045$; rods $t = 1.035$, $df = 3$, $p = 0.3768$; networks $t = 1.746$, $df = 3$, $p = 0.1791$; punctate mitochondria $t = 0.3717$, $df = 3$, $p = 0.7348$. UK5099 vs. DMSO: average area, paired t-test $t = 4.837$, $df = 6$ **$p = 0.0029$; rods Mann–Whitney test *$p = 0.0156$; networks paired t-test $t = 1.301$, $df = 6$, $p = 0.2409$; punctate mitochondria paired t-test $t = 2.987$, $df = 6$, *$p = 0.0244$). Data information: Data are reported as mean ± SE. Source data are available online for this figure.

in either total homogenate or the TIF were detected in neurons exposed to etomoxir or CB839 (Fig. 2B,C). These data support the idea that pyruvate entry into the mitochondria is required to ensure synapse formation during neuronal maturation.

## The inhibition of different metabolic pathways affects oxygen consumption rate and mitochondrial morphology in hippocampal neurons

Neuronal differentiation is strictly dependent on energy metabolism, and several lines of evidence have shown that mitochondrial activity controls synaptic plasticity and neuronal differentiation and maturation (Li et al, 2004; Pekkurnaz and Wang, 2022; Todorova and Blokland, 2017). Because neuronal maturation is strongly impaired when mitochondria MPC is inhibited (Figs. 1 and 2), we evaluated whether defects in neuronal structure were paralleled by alterations in mitochondrial activity and morphology (Fig. 3A). To determine whether this effect was specific to inhibition of MPC, we also assessed the effects of Gls-1 and CPT-1a inhibition.

As shown in Fig. 3B, neurons exposed to UK5099 displayed a significant deficit in oxygen consumption rate (OCR), leading to a decrease in basal, uncoupled, and maximal uncoupled respiration in comparison with the control condition. The same effect was elicited by the administration of CB839. On the other hand, etomoxir administration induced a significant increase in basal and uncoupled respiration compared to control, whereas no changes were observed in maximal uncoupled respiration (Fig. 3B). These observations suggest that glucose, glutamine, and fatty acid oxidation differentially contribute to neuronal respiration, with the first two pathways being the most involved in mitochondria energy supply.

OCR dysfunction might be accompanied by defects in mitochondrial morphology. To study mitochondrial morphology, hippocampal neurons were transfected with a plasmid expressing GFP-tagged subunit 9 of the F0-F1 ATPase to visualize mitochondria and a plasmid expressing the fluorescent protein mCherry to trace the neuronal structure (Fig. 3A). Analysis was performed along dendrites (Fig. EV2A), and different parameters, such as mitochondrial shape (rounded, rods, and network), number, and area, were considered. The inhibition of CPT-1a by etomoxir did not alter mitochondrial morphology, number, or area (Figs. 3C and

EV2B). On the other hand, CB839-induced Gls-1 inhibition resulted in the reduction of both the average area of mitochondria and rod length, without affecting other mitochondrial parameters (Figs. 3D and EV2C), indicating a reduction in mitochondrial density and complexity. MPC inhibition (UK5099 treatment) had the strongest effect on mitochondrial structure: we observed a reduction in the number of rounded punctate mitochondria that was accompanied by an increase in rod-shaped mitochondria and in the average length of rod and network branches in UK5099-treated neurons. Moreover, the average area of mitochondria increased with UK5099 treatment; in contrast, no changes were detected in junction, branch, or network numbers. These data suggest that inhibition of the shuttling of pyruvate into mitochondria led to increased mitochondrial mass and network complexity in differentiating neurons (Figs. 3E and EV2D). Together, these observations indicated that glucose oxidative metabolism, specifically the entry of pyruvate into the mitochondria, is a key event in determining proper mitochondrial function and shape in neuronal dendrites.

## MPC inhibition leads to decreased pyruvate transport into mitochondria that is associated with impaired glutamate synthesis

To investigate how the inhibition of energy pathways might affect the abundance of metabolic intermediates in differentiating neurons treated with the three different metabolic inhibitors, we performed a targeted metabolomic analysis by liquid chromatography coupled to tandem mass spectrometry (LC-MS/MS). As expected from the inhibition of fatty acid oxidation, our analysis showed that etomoxir reduced the intracellular levels of long-chain acylcarnitines, namely acylcarnitine-C18:1, acylcarnitine-C16:0, and acylcarnitine-C18:0, as well as glycine when compared to control (Fig. 4A). In contrast, we observed increased levels of the medium-chain acylcarnitine-C9 and oxaloacetate (OAA). In CB839-treated neurons, lower levels of lactate and alanine as well as αKG and the medium-chain acylcarnitine-C5 were detected (Fig. 4B). Of note, acetyl-CoA levels increased upon Gls-1 inhibition, suggesting that neurons preferentially rewired metabolism toward glucose, lactate and alanine utilization by promoting pyruvate oxidation in the mitochondria. This observation is corroborated by increased in mRNA levels of *Mpc1*, one of the

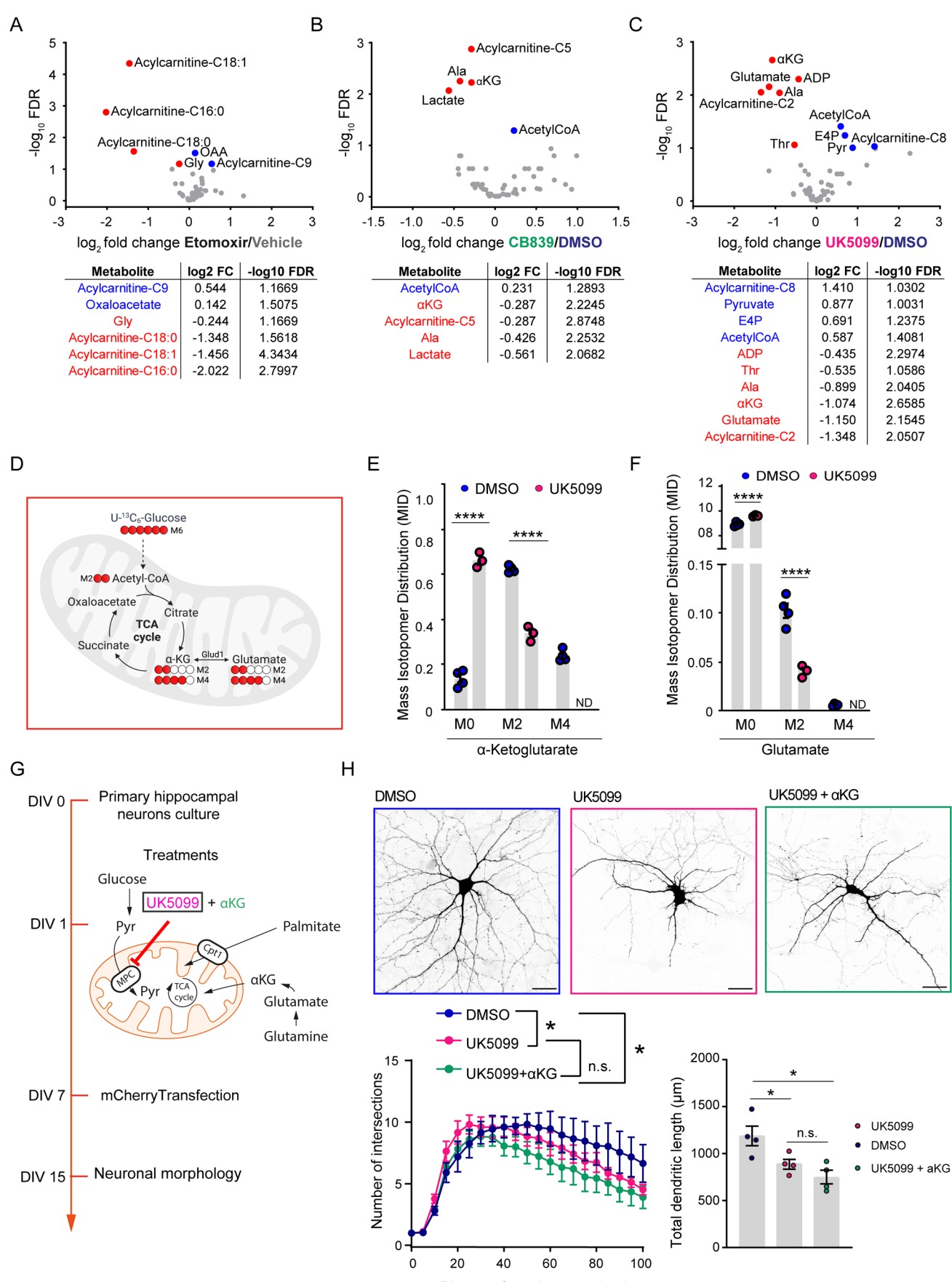

**Figure 4. Effect of Etomoxir, CB839, and UK5099 treatment on neuron metabolism and of α-ketoglutarate supplementation on neuronal complexity in UK5099-treated neurons.**

(A–C) Volcano plot of differentially regulated metabolites of DIV15 neurons treated with vehicle or the indicated metabolic inhibitors. (**A**) etomoxir, (**B**) CB839, and (**C**) UK5099. Tables report the significantly increased (blue) and decreased (red) metabolites. Statistical analysis was performed using one-way ANOVA followed by Fisher's least significant difference test. We corrected *p*-values with a Benjamini–Hochberg procedure to control the false discovery rate to adjust for multiple tests. (**D**) Schematic diagram of TCA Flux. (**E,F**) Mass isotopomer distribution (MID) of α-ketoglutarate (αKG) (**E**) and glutamate (**F**) labeled with [U-$^{13}$C$_6$] glucose. Data information: Data are presented as mean ± SE, $n = 4$ biological replicates for DMSO and $n = 3$ biological replicates for UK5099. Statistical analysis was performed using Multiple t-test, ****$p < 0.00001$ vs. DMSO. (**G**) Timeline showing the experimental procedure. Primary hippocampal neurons were isolated and plated (DIV0) and UK5099 alone or in the presence of 2.5 mM cell-permeable αKG was administered at DIV1. The transfection with a plasmid to express the mCherry protein was performed at DIV7 and analyses were performed at DIV15. (**H**) Representative confocal images (upper panel) of neurons transfected with mCherry and treated with DMSO and UK5099 alone or in the presence of cell-permeable αKG. Quantification of Sholl analysis and the total dendritic length are reported in the lower panels (scale bar 30 μm; Sholl analysis, $n = 4$ biological replicates; two-way ANOVA with Tukey's multiple comparison test; detailed information is reported Table EV1; dendritic length: $n = 4$ biological replicates; ≥20 neurons for each condition; RM one-way ANOVA with Tukey's multiple comparison test, UK5099 vs. DMSO *$p = 0.0256$, UK5099 + αKG vs. UK5099 $p = 0.2109$, and UK5099 + αKG vs. DMSO *$p = 0.0279$). Data information: Data are reported as mean ± SE. Source data are available online for this figure.

two subunits of the MPC, upon treatment with CB839, but not with UK5099 (Fig. EV3A). Importantly, glutamate levels were not affected by CB839 treatment (Fig. 4B), suggesting that glucose oxidation, lactate and alanine utilization are sufficient to maintain glutamate levels and that glutamine metabolism is mainly used by neurons to maintain mitochondrial functionality and structure (Figs. 3D and EV2C).

Strikingly, UK5099 increased pyruvate and acetyl-CoA levels, which were accompanied by reduced levels of short-chain acylcarnitine-C2, αKG, alanine, and glutamate (Fig. 4C). Upon UK5099 exposure, we also found lower levels of threonine and ADP, while medium-chain acylcarnitine-C8 and erythrose-4-phosphate were increased compared to control cells. These data demonstrated that MPC inhibition leads to a decrease in the critical metabolite αKG, an effect also observed with Gls-1 inhibition. Notably, only UK5099 treatment decreased glutamate levels and exhibited negative effects on neuronal terminal differentiation. Therefore, we hypothesize that glucose oxidation-derived glutamate might be an important metabolite in the neuronal maturation process.

## Pyruvate-derived glutamate is required for neuronal maturation and synaptic transmission

Glutamate is an essential amino acid that plays several roles in neurons, from being the major excitatory neurotransmitter, driving synaptic plasticity and transmission (Magi et al, 2019; Reiner and Levitz, 2018), to contributing to energy supply as a metabolic substrate. To investigate the contribution of glucose-derived carbons to glutamate, we labeled differentiating neurons treated with either DMSO or UK5099 with [U-$^{13}$C$_6$]-glucose and analyzed the levels of several isotopomers by LC-MS/MS (Fig. 4D). This analysis enables us to track the fate of labeled carbons ($^{13}$C) derived from glucose. Glucose initially undergoes glycolysis in the cytoplasm, producing M6 metabolites (indicating the number of labeled glucose-derived carbons) during the priming phase. At the end of this phase, aldolase breaks down fructose-1,6-bisphosphate into glyceraldehyde 3-phosphate and dihydroxyacetone phosphate leading to the generation of M3 metabolites (containing 3 carbons from labeled glucose). Subsequently, in the pay-off phase of glycolysis, M3 pyruvate is formed and transported into the mitochondria through MPC. Inside the mitochondrial matrix, M3 pyruvate undergoes decarboxylation by pyruvate dehydrogenase activity, resulting in the production of acetyl-CoA that retains only

two labeled carbons (M2) originating from [U-$^{13}$C$^6$]-glucose (M6). This M2 acetyl-CoA then enters the TCA cycle and gives rise to M2 αKG. Finally, the enzyme glutamate dehydrogenase 1 (Glud1), which is the major isoform present in the rat hippocampus, converts M2 αKG into M2 glutamate (Fig. 4D). In addition, a second round of the TCA cycle and the action of Glud1 on M4 αKG lead to the production of M4 glutamate (Fig. 4D). Metabolites labeled as M0 indicate that they are not derived from labeled glucose and are therefore unlabeled.

As expected, we did not observe any difference in M3 glyceraldehyde 3-phosphate/dihydroxyacetone phosphate (two isomers that cannot be distinguished by this type of analysis) or M3 phosphoenolpyruvate levels. These data suggest that glycolysis activity was unaffected by MPC inactivation. Moreover, we were not able to detect any M2 acetyl-CoA in UK5099-treated cells, although this was present in control cells (Fig. EV3B). In addition, we observed reduced levels of M2 and M4 αKG, indicating that glucose-derived carbons are necessary to sustain αKG intracellular levels (Fig. 4E). Accordingly, UK5099 treatment also significantly reduces M2 and M4 glutamate isotopomers (Fig. 4F). Furthermore, the levels of mRNA and protein of Glud1 were unaffected by UK5099 treatment (Appendix Fig. S2). Altogether these results support the notion that pyruvate oxidation in mitochondria is required for glutamate production in neurons.

Given that αKG produces glutamate by the action of transaminases and/or Glud1, we attempted to rescue the neuronal morphological phenotypes associated with MPC inhibition by treating differentiating neurons with either αKG or glutamate. To this end, we administered 2.5 mM cell-permeable αKG (dimethyl 2-oxoglutarate) (Betto et al, 2021) to UK5099-treated neurons at DIV1 (Fig. 4G). Assessment of neuronal morphology by Sholl analysis in mCherry-transfected neurons showed that αKG did not rescue the defects in neuronal arborization and dendritic length in UK5099-treated neurons (Fig. 4H). Nevertheless, it ameliorated the alterations in mitochondrial shape, number, and area (Fig. EV3C). Specifically, neurons treated with UK5099 in the presence of αKG showed a partial rescue of mitochondrial network numbers, and a full restoration of mitochondrial rod and of the average network branch length (Fig. EV3C). This indicates that αKG is important for mitochondrial function but is not a metabolite required for the proper development of the dendritic tree.

Next, we investigated whether glutamate supplementation in UK5099-treated neurons could rescue neuronal morphogenesis by treating primary hippocampal neurons with UK5099 and assessing

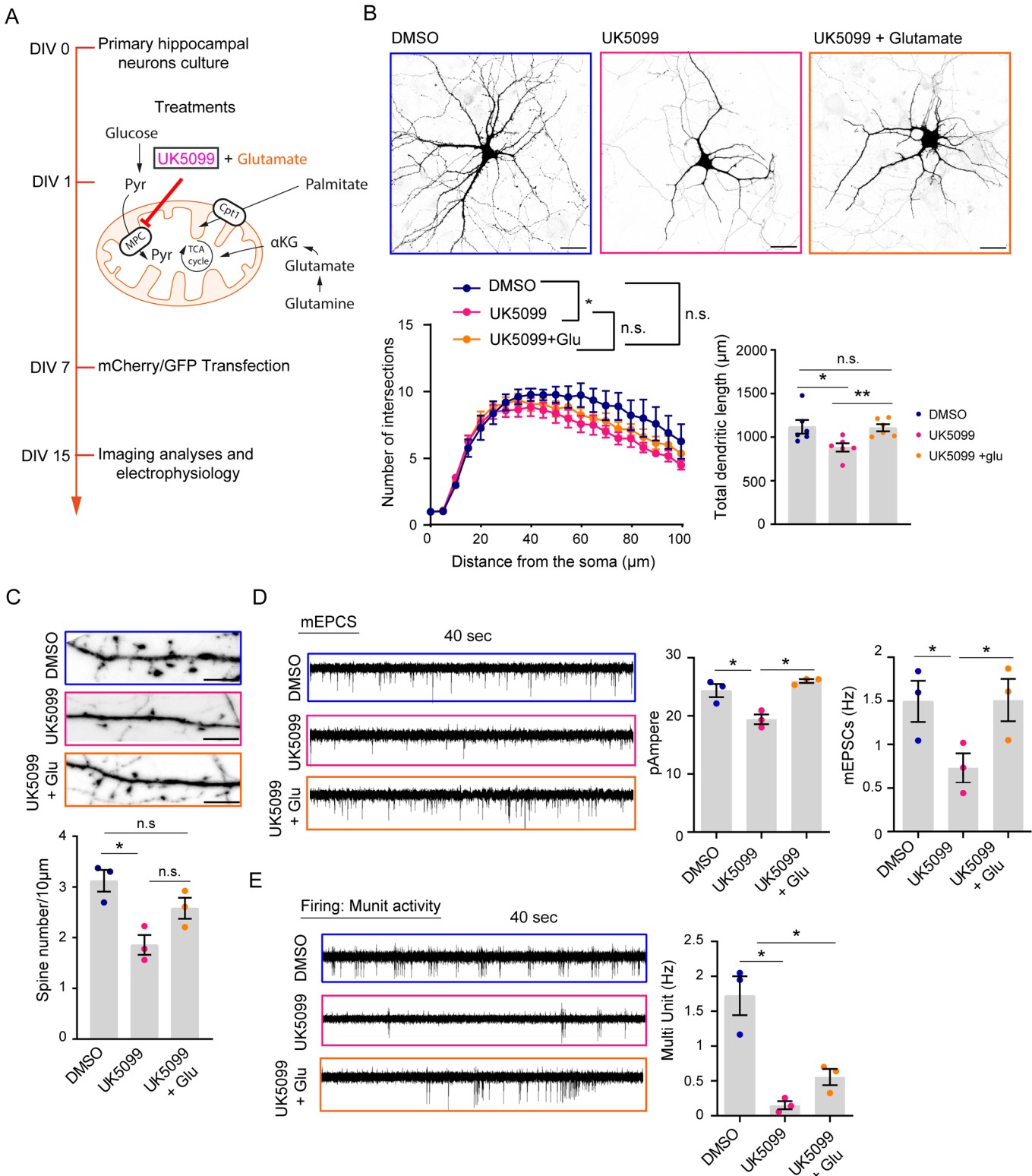

whether the presence of 10 μM glutamate (Serratto et al, 2020) could rescue the defects in neuronal maturation (Fig. 5A). First, we analyzed the glutamate levels in the culture media at different time points. At DIV 15, we found a reduced amount of glutamate in the medium of UK5099-treated neurons compared to control cells

($p = 0.073$, Fig. EV4A). These data underscore the requirement for secreted glutamate in the culture medium to sustain neuronal maturation in control cells. In addition, while glutamate levels were higher at the beginning of the rescue experiment (DIV3), they dropped to the levels detected in UK5099-treated cells by DIV7 and

**Figure 5. Glutamate partially rescues deficits in neuronal differentiation triggered by MPC inhibition.**

(A) Timeline of the experimental procedure. Primary hippocampal neurons were isolated and plated (DIV0) and UK5099 alone or in the presence of 10 µM glutamate was administered at DIV1. Transfection with a plasmid expressing the mCherry protein was performed at DIV7 and analysis was conducted at DIV15. (B) Representative confocal images (upper panel) of neurons transfected with mCherry and treated with DMSO and UK5099 alone or in the presence of glutamate. Quantification of Sholl analysis and the total dendritic length are reported in the lower panels (scale bar 30 µm; Sholl analysis, $n = 5$ biological replicates; two-way ANOVA with Tukey's multiple comparison test; detailed information is reported in Table EV2. Dendritic length, $n = 6$ biological replicates; ≥40 neurons for each condition; RM one-way ANOVA with Tukey's multiple comparison test, UK5099 vs. DMSO *$p = 0.0179$, UK5099 + Glu vs. DMSO $p = 0.9847$, and UK5099 + Glu vs. UK5099 **$p = 0.0087$). (C) Representative confocal images of dendritic spine density in neurons transfected with mCherry and quantification (scale bar 5 µm; 3 biological replicates; 19, 19, and 17 neurons for DMSO, UK5099, and UK5099 + Glu, respectively; RM one-way ANOVA uncorrected Fisher's LSD; UK5099 vs. DMSO *$p = 0.0357$, UK5099 + Glu vs. DMSO $p = 0.3061$, and UK5099 + Glu vs. UK5099 $p = 0.0834$). (D) Representative traces of miniature excitatory postsynaptic currents (mEPSCs) recorded in cultures treated with UK5099 alone (or DMSO) or with UK5099 and glutamate followed by quantification of frequency events and amplitude current (3 biological replicates; number of neurons: 18–20 [DMSO], 14–17 [UK5099], 15 [UK5099+ Glu]; mEPSCs amplitude statistic: RM one-way ANOVA followed by Tukey's multiple comparisons test: UK5099 vs DMSO *$p = 0.0178$, UK5099 + Glu vs. UK5099 *$p = 0.0149$, UK5099+ Glu vs. DMSO $p = 0.3085$; mEPSCs frequency statistic: RM one-way ANOVA uncorrected Fisher's LSD: UK5099 vs DMSO *$p = 0.0474$ UK5099 + Glu vs. UK5099 *$p = 0.0496$, UK5099+ Glu vs DMSO $p = 0.1967$). (E) Multi-unit (MU) activity recorded in neurons treated with UK5099 alone (or DMSO) or UK5099 and glutamate. Quantification of MU frequency in the right panel (3 biological replicates; number of neurons: 22 [DMSO], 14 [UK5099], and 24 [UK5099 + Glu]; MU frequency statistic: RM one-way ANOVA uncorrected Fisher's LSD: UK5099 vs. DMSO *$p = 0.04226$, UK5099 + Glu vs. UK5099 $p = 0.1442$, UK5099+ Glu vs. DMSO *$p = 0.0191$). Data information: Data are reported as mean ± SE. Source data are available online for this figure.

DIV10 (Fig. EV4A). Nevertheless, glutamate supplementation at DIV1 in UK5099-exposed neurons partially rescued dendritic complexity in terms of arborization and fully restored the total dendritic length in comparison with neurons treated with UK5099 alone (Fig. 5B). Of note, glutamate supplementation influenced spine density. As shown in Fig. 5C, the inhibition of MPC activity induced defects in spine density, while glutamate supplementation ameliorated this phenotype by promoting an increase in the density of dendritic spines compared to neurons treated with UK5099 alone.

We studied the functional impact of UK5099 treatment and glutamate supplementation in neurons by analyzing miniature postsynaptic currents. In line with the results described above, we detected a reduction in the amplitude of excitatory events (mEPSCs), which was fully normalized by early glutamate delivery (Fig. 5D). Analysis of mEPSCs also highlighted a lower frequency of events in UK5099-treated neurons, suggesting a reduced number of pre-synaptic terminals, which was also rescued by glutamate administration (Fig. 5D). Finally, we monitored changes in the spontaneous cell firing activity via voltage clamp recordings in the cell-attached modality. As indicated in Fig. 5E, MPC inhibition resulted in a significantly lower ability to generate action potentials, which showed a tendence to increase in glutamate-supplemented neurons.

We then asked which potential mechanisms underlie this effect. We wondered whether a rescue in mitochondrial function could be implicated in the glutamate-driven recovery of the neuronal terminal differentiation process in UK5099-treated cells. We assessed the effect of glutamate on mitochondrial OCR in neurons treated with UK5099. Our analysis showed that the administration of glutamate did not offset the strong deficits induced by MPC inhibition on the basal OCR and on the uncoupled and maximal respiration in primary hippocampal neurons (Fig. EV4B). In line with this result, we observed no changes in mitochondrial shape, area, and number. In fact, no major changes were found in the mitochondrial morphology of neurons treated with UK5099 in the presence of glutamate compared with neurons exposed only to UK5099. A partial compensation was detected in the rod, network, and junction numbers, but not in the other parameters analyzed (Fig. EV4C). These results indicate that supplementation with glutamate does not restore the mitochondrial defects caused by the blunted MPC activity. Therefore, although pyruvate-derived

glutamate is not sufficient to sustain mitochondrial activity in neurons, it is essential for maturation and dendritic arborization.

## Mpc1 genetic knockdown recapitulates key features of MPC chemical inhibition

Given that blunting MPC activity deeply affects neuronal structural, electrical, and metabolic complexity, we evaluated the impact of the genetic downregulation of MPC subunit Mpc1. First, we analyzed, glutamate and αKG levels in mice with a specific deletion of Mpc1 in glutamatergic neurons. To this aim, we took advantage of neuro-Mpc1-KO mice generated by De La Rossa collaborators (De La Rossa et al, 2022). To generate neuro-Mpc1-KO mice, Mpc1 flox/flox mice were crossed with the commercially available CamkIIaCreERT2 mice. To achieve deletion of Mpc1 floxed alleles in the CamKIIα-expressing adult neurons, neuro-Mpc1-KO mice were treated with Tamoxifen as previously described by De La Rossa A et al (De La Rossa et al, 2022). Metabolomic analysis of hippocampal samples showed significantly reduced levels of glutamate in neuro-Mpc1-KO mice compared to the control animals (neuro-Mpc1-WT mice), while αKG showed a similar trend which does not reach statistical significance (Fig. 6A). Prompt by these results that recapitulate the metabolic features observed with the MPC chemical inhibition, we transduced primary hippocampal neurons with a viral vector expressing a short harpin RNA targeting Mpc1 sequence [Sh Mpc1 (A)] or a scramble sequence [SCR (A)] at DIV1 (Fig. 6B). Mpc1 silencing resulted in downregulation of the protein at DIV15 (Fig. 6C). We thus performed the same morphological and metabolic analysis as for the MPC chemical inhibition. Importantly, we observed the same phenotypes described for the exposure to UK5099. Mpc1 silencing led to a significant decrease in PSD-95 levels (Fig. 6C) and in a reduction of dendritic arborization and of the total dendritic length compared to the control condition (Fig. 6D). Furthermore, tracing experiments performed in Mpc1-silenced neurons and labeled with [U-$^{13}C_6$]-glucose showed a significant decrease in glutamate M2 and M4 isotopomers compared to scramble control neurons (Fig. 6E). Finally, to further strengthen our results, we used another cellular model (i.e., primary cortical neurons) and a previously validated shRNA sequence targeting Mpc1 [Sh Mpc1 (B)] (De La Rossa et al, 2022) (Appendix Fig. S3A). The results of metabolomic analyses showed statistically significant

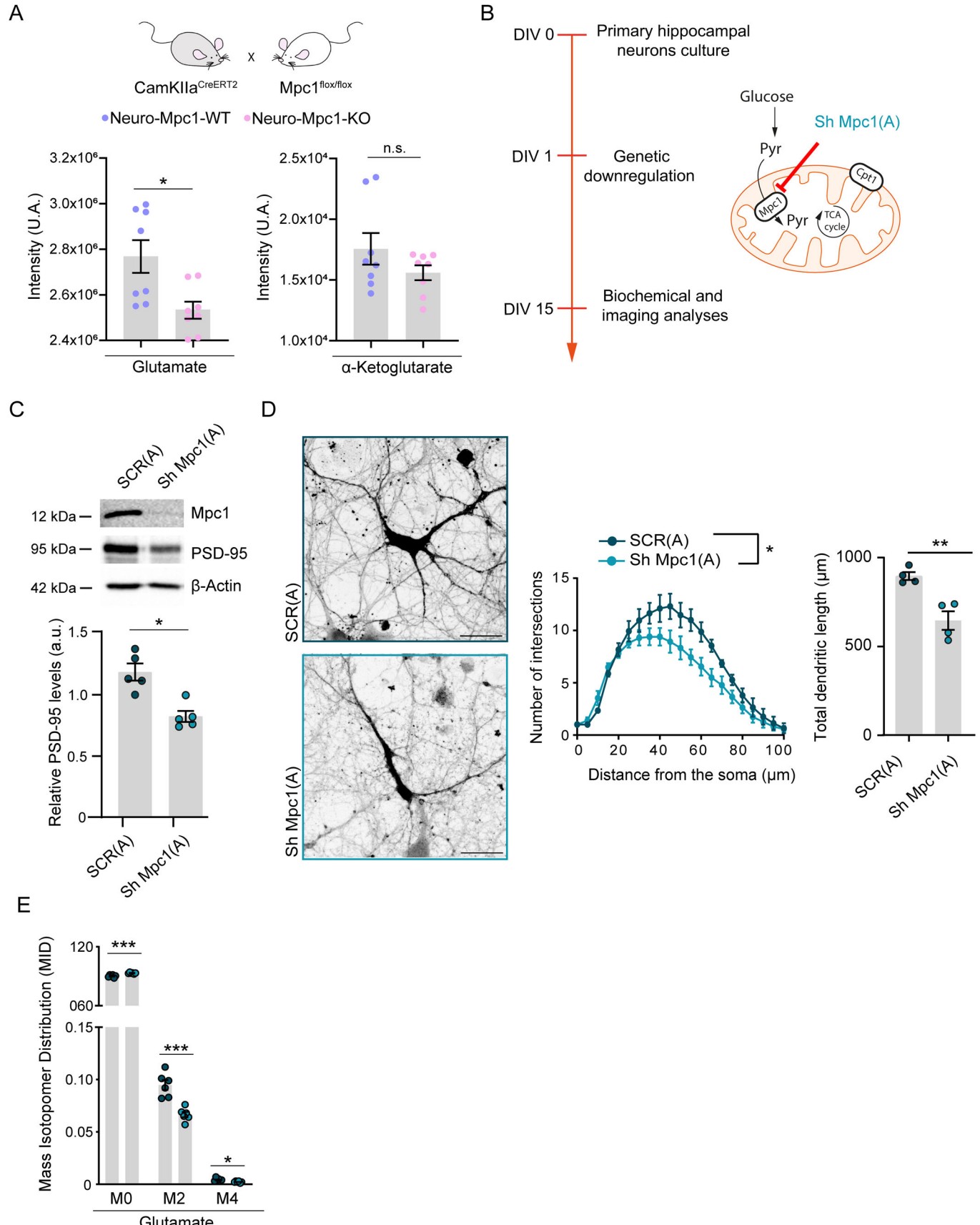

Figure 6.  **Genetic deletion of Mpc1 affects glutamate levels and neuronal differentiation.**

(A) The graphs on the bottom show the steady-state abundance of glutamate (left) and α-ketoglutarate (right) in the hippocampus isolated from adult Neuro-Mpc1-WT and -KO mice, determined by untargeted metabolomics ($n = 8$ independent mice per condition, mean ± SE. For glutamate Neuro-Mpc1-WT: 2768286 ± 71560 A.U; Neuro-Mpc1-KO: 2532304 ± 37287 A.U. *$p = 0.011$, Unpaired t test. For αKG Neuro-Mpc1-WT: 17562 ± 1302 A.U; Neuro-Mpc1-KO: 15596 ± 1737 A.U. $p = 0.194$, n.s., Unpaired t test). (B) Timeline of the experimental procedure. Primary hippocampal neurons were isolated and plated (DIV0) and infected with a viral vector expressing the eGFP and a shRNA sequence to downregulate Mpc 1 [Sh Mpc1 (A)] or with a control sequence [(SCR(A)] at DIV1. Analysis was conducted at DIV15. (C) Representative western blot for Mpc1 and PSD-95 (upper panel) and quantification of PSD-95 (lower panel) in the homogenate of neurons silenced for Mpc1 compared to the SCR ($n = 5$ biological replicates, paired t-test, t = 4.094, df = 4 *$p = 0.0149$; mean ± SE). (D) Representative confocal images (left panels) of hippocampal neurons silenced for Mpc1 compared to the SCR (scale bar 30 μm). The quantifications of the dendritic arborization (Sholl analysis) and total dendritic length are reported in the graphs on the right. (5 biological replicates: SCR (A): 38 neurons, Sh Mpc1 (A): 40 neurons. For Sholl analysis: Multiple t-test, degrees of freedom (df) = 8, t = 2.750, *$p = 0.025$, t = 2.626 *$p = 0.030$, t = 2.988, *$p = 0.0174$ at 55, 60, and 65 μm from the soma, respectively. For dendritic length: 4 biological replicates, Paired t-test, t = 6.872, df = 3, **$p = 0.0063$; mean ± SE). (E) Mass isotopomer distribution (MID) of glutamate labeled with [U-$^{13}$C$_6$] glucose ($n = 6$ biological replicates; Multiple t-test, M0 ***$p = 0.00041$, M2 ***$p = 0.00035$, M4 *$p = 0.02$; mean ± SE). Source data are available online for this figure.

reduction of glutamate levels in Mpc1-silenced neurons compared to control cultures (Appendix Fig. S3B). These data corroborate our results supporting the importance of MPC function for glutamate production and neuronal maturation.

## A glucose–pyruvate–glutamate axis sustains local protein synthesis at synapses

Given the partial recovery of spine density induced by glutamate supplementation, we analyzed PSD-95 protein and mRNA levels. The results of western blot analysis (Fig. 7A) and quantitative RT-PCR (Fig. 7B) revealed that glutamate did not rescue the expression of PSD-95 in UK5099-treated neurons at either the mRNA or the protein level. Of note, UK5099 treatment reduced PSD-95 protein levels without affecting mRNA expression (Fig. 7A,B), suggesting that the inhibition of MPC function mainly affected PSD-95 translation rather than transcription.

This led us to evaluate global protein synthesis with SUrface SEnsing of Translation (SUnSET) technique (Schmidt et al, 2009), a nonradioactive method to detect translational rate in live cells. Briefly, we fed primary hippocampal neurons with 2 μM puromycin, which is randomly incorporated in the nascent polypeptide chain, and lysed them in Laemmli buffer. Puromycin incorporation was revealed by western blot. As shown in Fig. 7C, in neurons treated with UK5099 in either the absence or presence of glutamate, the global translational rate is drastically reduced compared to control neurons. These data indicate that UK5099 impairs global protein synthesis, and that glutamate supplementation is not able to ameliorate this effect.

However, it is widely accepted that control of protein synthesis plays a critical role at the synaptic compartment in neurons. Indeed, the translation of synaptic transcripts in defined spatio-temporal windows orchestrates synaptic processes, such as synapse maturation and synaptic plasticity (Holt et al, 2019; Rangaraju et al, 2017; Hafner et al, 2018). We thus investigated protein synthesis rate at the level of synapses by isolating synaptoneurosomes from primary hippocampal neurons (Appendix Fig. S4), treated with UK5099 alone or in combination with glutamate, and performing a SUnSET assay using biochemical and imaging approaches. Notably, western blot analysis showed that the effect of UK5099 was evident even at the synapse, as the levels of protein synthesis were significantly decreased in synaptoneurosomes (Fig. 7D). Nonetheless, the supplementation with glutamate induced a partial rescue of the translational rate in the synaptoneurosomes purified from neurons exposed to UK5099 (Fig. 7D). To confirm these

results, we performed a SUnSET assay using the same conditions and analyzed puromycin levels at dendritic spines by confocal imaging in GFP-transfected neurons. When MPC activity was blunted by UK5099 treatment, the dendritic spines exhibited a low translational rate, while the addition of glutamate partially rescued the protein synthesis levels (Fig. 7E), suggesting that the glucose–pyruvate–glutamate metabolic axis triggers local protein synthesis in neurons. Because PSD-95 protein is critical for modulating synaptic function (Won et al, 2017; Coley and Gao, 2018) and its transcript is locally translated (Donlin-Asp et al, 2021; Ifrim et al, 2015), we examined PSD-95 protein levels in synaptoneurosomes isolated from primary neurons exposed to UK5099 alone or in the presence of glutamate. Consistent with our previous observations (Fig. 2B,C), upon MPC inhibition, PSD-95 expression decreased compared to the control. However, the administration of glutamate to UK5099-exposed neurons fully normalized PSD-95 protein levels at synapses (Fig. 7F), indicating that glucose-derived glutamate is a key metabolite required for the synthesis of synaptic proteins.

Glutamate can activate different receptors and trigger several signaling pathways that can affect protein translation rate in a different manner. For instance, the activation of N-methyl-D-aspartic acid receptor attenuates protein synthesis at the synapses (Marin et al, 1997; Scheetz et al, 2000; Luchelli et al, 2015), while group I mGluRs signaling stimulates local protein translation in neurons (Lüscher and Huber, 2010; Costa-Mattioli et al, 2009). In light of this consideration, we investigated whether selective activation of group I mGluRs in UK5099-treated neurons could rescue the defects in protein translation. We treated UK5099-exposed primary hippocampal neurons with (S)-3,5-Dihydroxy-phenylglycine (DHPG), a specific agonist of group I mGluRs, and we analyzed global and local protein translation (Fig. 8A).

The results of SUnSET assay showed that DHPG did not affect the reduction of the global translational rate in UK5099-exposed neurons (Fig. EV5A), as reported above in experiments testing the effects of glutamate supplementation (Fig. 7C). On the other hand, the presence of DHPG fully restored local translation and normalized PSD-95 protein levels at synapses (Fig. 8B). As proof of concept of mGluRs signaling involvement in neuronal maturation, we exposed neuronal cultures at DIV1 to 2-Methyl-6-(phenylethynyl)pyridine (MPEP), an antagonist of mGluRs (Gasparini et al, 1999). The inhibition of mGluR phenocopied the effects of UK5099 on global protein translation and on PSD-95 protein levels (Fig. EV5B), thus supporting the implication of mGluRs in the mechanisms underlying neuronal maturation.

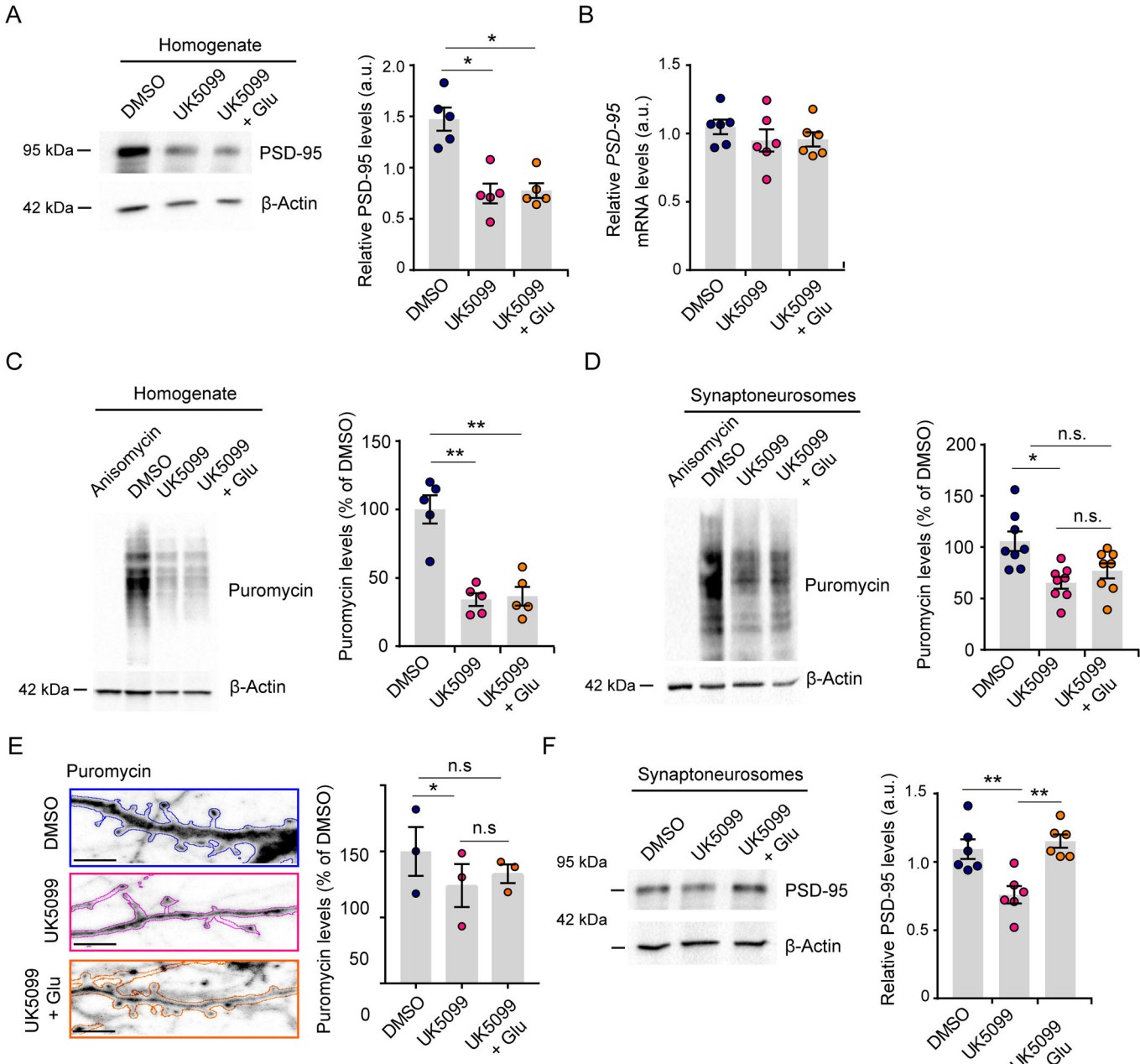

To assess the effects of mGluRs activation on neuronal architecture, we analyzed dendritic arborization and spine maturation. The Sholl analysis showed that UK5099-treated neurons fully rescued dendritic complexity and dendritic length upon DHPG exposure (Fig. 8C). Spine analysis revealed that DHPG treatment completely restored spine density in UK5099-exposed neurons. The examination of the different spines categories showed that the different spine populations were associated with treatments (Fig. 8D). UK5099 treatment significantly reduced the percentage of mature mushroom spines and increased the immature thin spines and filopodia, while DHPG restored physiological spine maturation (Fig. 8D). These data suggest that mGluRs signaling is critical to recover the impairment of local protein translation and, thereby, restores neuronal maturation due to the reduced generation of glucose-derived glutamate.

Taken together, our data highlight the importance of energy metabolism and the biosynthesis of glucose metabolic intermediates for neuronal maturation and function.

## Discussion

Current knowledge about the role of key metabolic pathways in neuronal maturation has not been fully elucidated yet. It is widely accepted that the brain uses glucose primarily as an energy source, however, other key metabolites could be produced from glucose metabolism to enable proper neuronal maturation. Our data

**Figure 7. Local translation at synapses requires glucose-derived glutamate.**

Neurons were treated with UK5099 alone or in combination with glutamate according to the scheme reported in Fig. 5A and analyzed as follows. (**A**) Representative western blot and quantification of PSD-95 protein in the homogenate, normalized to β-Actin ($n = 5$ biological replicates, repeated measures [RM] one-way ANOVA with Tukey's multiple comparison test $F_{(1.649, 6.598)} = 12.32$; UK5099 vs. DMSO *$p = 0.0452$, UK5099 + Glu vs. DMSO *$p = 0.0277$, UK5099+ Glu vs UK5099 $p = 0.975$). (**B**) RT-qPCR of *PSD95* mRNA, normalized to the mRNA encoding α-tubulin and *histone H3* ($n = 6$ biological replicates, RM one-way ANOVA, Tukey's multiple comparison test $F_{(1.793, 8.964)} = 0.515$; UK5099 vs. DMSO $p = 0.691$, UK5099 + Glu vs. DMSO $p = 0.5909$, UK5099 + Glu vs. UK5099 $p = 0.9973$). (**C,D**) Representative western blots and quantification of puromycin incorporation in total extract (**C**) and synaptoneurosomes (**D**). Anisomycin was used as a negative control and β-actin was used as a normalizer ($n = 5$ biological replicates and $n = 8$ biological replicates for homogenate (**C**) and synaptoneurosomes (**D**), respectively; RM one-way ANOVA with Tukey's multiple comparison test; Homogenate: $F_{(1.234, 4.935)} = 36.44$, UK5099 vs. DMSO **$p = 0.0071$, UK5099 + Glu vs. DMSO **$p = 0.0075$, UK5099 + Glu vs. UK5099 $p = 0.8301$; Synaptoneurosomes: $F_{(1.307, 9.152)} = 6.41$, UK5099 vs. DMSO *$p = 0.0299$, UK5099 + Glu vs. DMSO $p = 0.1908$, UK5099 + Glu vs. UK5099 $p = 0.2497$). (**E**) Representative confocal images of puromycin immunolabeling in EGFP-transfected neurons (outline) and quantification of puromycin incorporation in dendritic spines (scale bar 2 µm; 3 biological replicates, 16, 19, and 17 neurons for DMSO, UK5099, and UK5099 + Glu, respectively; RM one-way ANOVA with Tukey's multiple comparison test; UK5099 vs. DMSO *$p = 0.0407$, UK5099 + Glu vs. DMSO $p = 0.4823$, UK5099 + Glu vs. UK5099 $p = 0.6635$). (**F**) Representative western blot and quantification of PSD-95 levels in synaptoneurosomes ($n = 6$ biological replicates, ordinary one-way ANOVA with Tukey's multiple comparison test; $F_{(2,15)} = 11.56$, UK5099 vs. DMSO **$p = 0.0047$, UK5099 + Glu vs. DMSO $p = 0.7892$, UK5099 + Glu vs. UK5099 **$p = 0.0013$). Data information: Data are reported as mean ± SE. Source data are available online for this figure.

strongly support the concept that glucose needs to be oxidized in the mitochondria to sustain both αKG production to ensure proper mitochondrial shape and network structure and glutamate production to support neuronal maturation and synaptogenesis.

Combining morphological, functional, and metabolic assays, we demonstrate the following: (i) MPC chemical or genetic inhibition dramatically impairs neuronal architecture, spine density, and protein translation; (ii) the effects of reduced activity of MPC demonstrate that glutamate levels mainly depend on glucose oxidation; (iii) glutamate supplementation restores dendritic length and synaptic transmission and partially rescues spine density by fostering local translation of synaptic proteins, such as PSD-95; (iv) activation of group I mGluR is sufficient to rescue local protein translation and, thereby, the impairment of neuron structure and spine morphology driven by MPC inhibition.

The goal of our study was to investigate the contribution of different metabolic pathways to the neuronal maturation process. To achieve this goal, we used primary hippocampal neuronal cultures as an in vitro terminal differentiation model to perform experiments mainly on glutamatergic neurons (Dotti et al, 1988). It is well recognized that glutamate-mediated neurotransmission requires a large amount of energy expenditure, consuming up to 80% of the energy expended in the gray matter (Sibson et al, 1998; Attwell and Laughlin, 2001; Shulman et al, 2004; Hyder et al, 2006). Therefore, there is a close connection between glutamatergic neurotransmission, energy demand, and nutrient utilization.

We selectively inhibited fatty acid β-oxidation, complete oxidation of the glucose, and glutaminase activity to assess their impact on mitochondrial metabolism and function and neuronal morphological changes. Etomoxir, the Cpt-1a inhibitor, increased mitochondrial functionality, likely by fostering glucose and/or glutamine utilization. Moreover, neurons did not show any structural and/or morphological changes upon etomoxir treatment, suggesting that fatty acid β-oxidation does not play a major role in the maturation process in our experimental model. On the other hand, and in line with previous results (Velletri et al, 2013; Agostini et al, 2016), glutaminase inhibition negatively affected mitochondrial function and network as well as neuronal terminal differentiation, which was limited to dendritic length in our study.

The most striking results were obtained with pharmacological inhibition of the mitochondrial pyruvate transporter MPC and further corroborated by genetic loss of function approaches. Taking advantage of both genetic and pharmacological tools, De La Rossa and collaborators demonstrated that reduced MPC activity leads to a decrease in OCR and ATP production in glutamatergic neurons (De La Rossa et al, 2022). In line with these results, we observed a reduced OCR and an altered mitochondrial network associated with a dramatic reduction in neuronal dendritic length and arborization and a decrease in spine density.

From a morphological point of view, inhibition of either MPC or Gls-1 activity led to a phenotype of reduced neurite length. However, only the chemical and genetic inhibition of MPC was able to alter both the arborization of neurites and spine density.

These data are in line with the results recently published by Iwata and colleagues showing that the exposure to UK5099 at DIV3 decreased dendritic growth and complexity in primary mouse neurons (Iwata et al, 2023).

It has been shown that mitochondrial pyruvate shuttling from cytoplasm is critical in the neurogenesis process because it controls neural stem/progenitor cells activation (Petrelli et al, 2023). However, most of our experiments were performed on a model of terminal neuronal maturation, a later stage of the neuronal development process. On this ground, the discrepancies found by other colleagues in the field in terms of dendritic development, spine formation, and mitochondrial morphology are probably due to the use of models of the early stages of neurogenesis ranging from stem/progenitor cells proliferation to fate acquisition and neuronal differentiation (Petrelli et al, 2023; Iwata et al, 2020).

Metabolomic data revealed that the MPC inhibitor (UK5099) and Gls-1 inhibitor (CB839) decreased αKG levels, but only UK5099 treatment was able to reduce glutamate levels; these latter data were also recapitulated in the hippocampus of neuro-Mpc1-KO mice and in Mpc1 silenced hippocampal and cortical neurons. Based on this, we attempted to rescue the phenotype induced by MPC inhibition by providing αKG or glutamate in the medium. The results showed that treatment with αKG was unable to rescue the arborization of neurons or the length of neurites. Conversely, glutamate treatment was effective in recovering the ability to generate action potentials, normalizing neurite length, and partially rescuing neurite arborization and spine density.

One of the main features of cell metabolism is its capacity for plasticity (Folmes et al, 2012). Indeed, when a metabolic route is chemically or genetically inhibited, metabolism is rearranged, and the lack of a key metabolite is thereby compensated through the

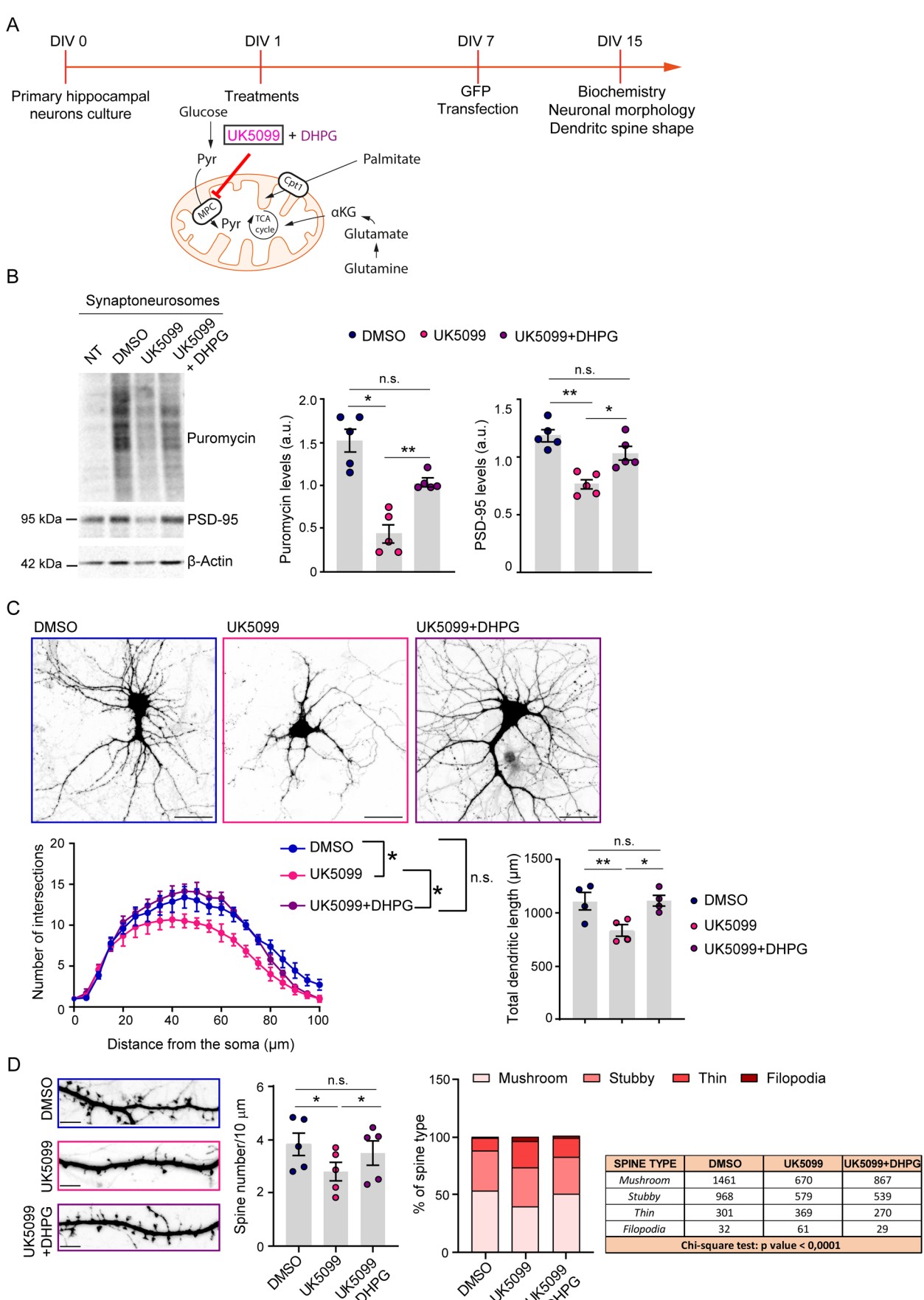

**Figure 8.   DHPG supplementation restores synaptic protein synthesis, dendritic complexity, and spine morphology in UK5099 neurons.**

(A) Timeline of the experimental procedure. Primary hippocampal neurons were isolated and plated (DIV0) and UK5099 alone or in the presence of 50 µM DHPG was administered at DIV1. Transfection with a plasmid expressing the GFP protein was performed at DIV7 and analysis was conducted at DIV15. (B) Representative western blot and quantification of puromycin incorporation and PSD-95 expression in synaptoneurosomes isolated from with UK5099 in presence/absence of DHPG. β-Actin was used as loading control and normalizer. Data information: Data are shown as mean ± SE ($n = 5$ biological replicates. For SUnSET: Ordinary one-way ANOVA, Tukey's multiple comparison test UK5099 vs DMSO *$p = 0.0233$; UK5099 + DHPG vs UK5099 **$p = 0.0071$, UK5099 + DHPG vs DMSO $p = 0.0936$. For PSD-95 expression: Ordinary one-way ANOVA uncorrected Fisher's LSD UK5099 vs DMSO **$p = 0.0042$; UK5099 + DHPG vs UK5099 *$p = 0.0430$, UK5099 + DHPG vs DMSO $p = 0.2138$). (C) Representative confocal images are shown in the upper panels (scale bar 30 µm). The quantifications of the dendritic arborization (Sholl analysis) and total dendritic length are reported in the lower graphs. (For Sholl analysis $n = 4$ biological replicates; DMSO 29 neurons, UK5099 32 neurons; UK5099 + DHPG 29 neurons; two-way ANOVA with Tukey's multiple comparison test; detailed information is reported in Table EV3. For dendritic length $n = 4$ biological replicates; DMSO 31 neurons, UK5099 28 neurons, UK5099 + DHPG 25 neurons; RM one-way ANOVA followed by Tukey's multiple comparison test ; UK5099 vs DMSO **$p = 0.0072$; UK5099 + DHPG vs DMSO $p = 0.9866$; UK5099 + DHPG vs UK5099 *$p = 0.0424$). Data information: Data are reported as mean ± SE. (D) Representative confocal images are reported on the left (scale bar 5 µm). In the middle graph quantification of spine number/10 µm of dendrites is shown for each condition (Data information: Data are shown as mean ± SE; $n$: 29; 34 and 28 neurons for DMSO, UK5099 and UK5099 + DHPG, respectively, from 5 biological replicates. RM one-way ANOVA Uncorrected Fisher's LSD; UK5099 vs DMSO *$p = 0.0338$; UK5099 + DHPG vs UK5099 *$p = 0.0423$, UK5099 + DHPG vs DMSO $p = 0.1536$. The stacked bar chart on the right shows the percentage of the different types of spines in DMSO, UK5099, and UK5099 in presence of DHPG-treated neurons. The joint distribution of dendritic spine type and treatment is represented in the table together with the $p$ value of the chi-square test of independence (chi-square test ****$p$ value < 0.0001). Source data are available online for this figure.

action of alternative pathways. As an example, although pyruvate is synthesized by different metabolic pathways (e.g., glycolysis, transamination), it can be shuttled into the mitochondria only by MPC. Therefore, due to metabolic plasticity, the effects of MPC inhibition can be compensated by other metabolic pathways (Zangari et al, 2020). Astrocyte-derived lactate acts as an additional energy source for neurons, as it can be converted into pyruvate, as elegantly described by Bonvento and Bolaños (Bonvento and Bolaños, 2021). In Gls-1 inhibited neuronal cultures, we noticed reduced levels of two pyruvate precursors, lactate and alanine accompanied by elevated levels of acetyl-CoA and *Mpc1* mRNA. These findings indicate that pyruvate-glutamate levels are maintained to support neuron maturation. Conversely, inhibiting MPC prevents neurons from compensating for the reduced glutamate generation within the mitochondria. In summary, our results suggest that inhibiting glutamine metabolism, which has a milder impact on neuronal morphology compared to MPC inhibition, triggers the generation of sufficient glutamate using alternative pyruvate-generating molecules such as glucose, lactate, and alanine. Consequently, this glutamate is released outside the cells and functions as a neurotransmitter. This represents a remarkable example of metabolic plasticity. Notably, in cells with MPC inhibition, the attempts of glutamine metabolism to compensate for the absence of glucose-derived glutamate were unsuccessful, leading to the observed detrimental phenotype of neurons.

We also demonstrated that glutamate supplemented at the initial phase of differentiation (DIV1) acted mainly as a neurotransmitter. We showed that the activation of the group I mGluRs is sufficient to fully recover the neuronal structure and spine morphology defects triggered by MPC inhibition. Mechanistically, glutamate activates mGluRs and, thereby, promotes local synaptic translation, the key mechanism that ensures proper synaptic maturation, plasticity, and transmission through a spatiotemporally defined modulation of the proteome at synapses (Fernandez-moya et al, 2014; Holt et al, 2019; Rangaraju et al, 2017). This effect on local translation accounts for the rescue of the morphological phenotype and synaptic function detected in mature neurons. In addition, glutamate was unable to normalize compromised mitochondrial function in MPC-inhibited neurons, while αKG supplementation improved mitochondrial shape and network. These data suggest that MPC activity orchestrates proper mitochondrial function by fueling the TCA cycle, as demonstrated

by αKG supplementation experiments, thus enabling glutamate synthesis and release, and ultimately favoring its role as a neurotransmitter more than as an energetic substrate (Divakaruni et al, 2017). This latter statement is evident from experiments in which exogenous glutamate was administered in MPC-blunted neurons. The canonical model predicts that the glutamine supplied to neurons by astrocytes is converted into glutamate and stored in synaptic vesicles ready for release (Bélanger et al, 2011). Here, we strikingly showed that glutamate can also be endogenously produced from the glucose–pyruvate–glutamate axis in neuronal mitochondria. Our data support the concept that in our neuronal model, MPC is an obligate bottleneck for mitochondrial generation of glutamate. Notably, MPC inhibition has been exploited in excitotoxic conditions typical of neurodegenerative disorders to reduce endogenously derived glutamate (Shah et al, 2014; Ghosh et al, 2016; Divakaruni et al, 2017). In addition, MPC deficiency due to pathogenic *MPC1* gene variants is a rare autosomal recessive disorder involving developmental delay and microcephaly (Jiang et al, 2022; Oonthonpan et al, 2019; Brivet et al, 2003). These data support the relevance of MPC during neurodevelopment.

Most of the present work was conducted on an in vitro model of neuronal terminal differentiation. We are aware that additional in vivo studies are necessary to further corroborate our findings to pave the way to the design of novel therapeutic strategies targeting MPC for the treatment of neurodevelopmental disorders.

In summary, this study demonstrates for the first time to the best of our knowledge that the glucose–pyruvate–glutamate axis is the main metabolic pathway enabling the glutamate production required during neuronal maturation.

## Methods

### Animal care

All procedures involving the preparation of primary hippocampal cultures were conducted in accordance with the ethical standards of the Institutional Animal Care and Use Committee of the University of Milan (Italian Ministry of Health permit #5247B.N.YCK/2018). Animals were maintained in cages with free access to food and water in a controlled environment with a 12 h/12 h light/dark cycle and a controlled temperature of 22 °C.

The neuro-Mpc1-KO mice in which MPC1 was knocked out specifically in excitatory glutamatergic neurons were obtained as previously described (De La Rossa et al, 2022). The neuro-Mpc1-KO phenotypes were tamoxifen-inducible. To induce Mpc1 deletion, the mice were injected intraperitoneally for 5 consecutive days with 100 μl of 10 mg/ml tamoxifen (Sigma, 85256) in sunflower oil. The mice were considered to be Mpc1 KO from 1 week after the final injection. All experiments were carried out in accordance with the Institutional Animal Care and Use Committee of the University of Geneva and with permission of the Geneva cantonal authorities (Authorization numbers GE/42/17, GE/70/15, GE/123/16, GE/86/16, GE/77/18, GE/205/17) and of the Veterinary Office Committee for Animal Experimentation of Canton Vaud (Authorization number VD3081).

## Primary neurons and treatments

Hippocampal primary neurons were isolated from the rat hippocampus at embryonic day 18–19 (E18-19) as previously described (Pelucchi et al, 2020). At days in vitro (DIV) 1, neurons were treated with 20 μM etomoxir, 4 μM CB839, or 50 μM UK5099. CB839 and UK5099 were dissolved in DMSO while etomoxir in water. DMSO- and water-treated neurons were used as controls. For rescue experiments, 10 μM glutamate or 2.5 mM DM-αKG (dimethyl 2-oxoglutarate or a cell permeable form of α-ketoglutarate, Sigma-Aldrich, cat. 349631-5G) were added to DIV1 neurons in the presence of UK5099. Compounds were left in the cell medium for the duration of the experiment (until DIV15). Treatment with 50 μM DHPG ((RS)-3,5-Dihydroxyphenylglycine, HELLOBIO, cat. HB0026) was performed at DIV1 in presence of UK5099. Treatment with 50 μM MPEP hydrochloride (Tocris, cat. 1212) was performed at DIV1. When reported, transfection with calcium phosphate was performed at DIV7 and biochemical and morphological analyses were conducted at DIV15. Neuronal cultures were transfected with the following plasmids: (1) the plasmid pcDNA3.1 containing subunit 9 of the F0-F1 ATPase tagged with GFP to label mitochondria; (2) the vector pAAV-U6>CaMKIIprom>mcherry, which expresses the mCherry fluorescent protein under the control of the neuron-specific promoter CaMKII and was used as a neurospecific filler; and (3) an EGFP expression vector, which was a gift from Dr. Maria Passafaro (CNR, Milan, Italy).

To downregulate Mpc1, in Biosafety Level 1 laboratory primary neurons were transduced with an adeno-associated viral vector expressing a shorth harpin RNA targeting Mpc1 [ShMpc1 (A): pAAV[ShRNA]-EGFP-U6 > rMPC1; target sequence ACACG-GAAGACTATCTTTAAT) or with a scramble sequence (SCR (A): pAAV[ShRNA]-EGFP-U6>Scramble_shRNA; target sequence CCTAAGGTTAAGTCGCCCTCG). Both viral vectors express the GFP protein under the control of the CMV promoter as reporter gene. The viral vectors were purchased from Vector Builder.

For the preparation of primary cortical cultures, wild-type pregnant mice were decapitated and E18 embryos were collected in HBSS medium. Primary cultures of cortical neurons were prepared as described previously (Fauré et al, 2006). Briefly, cortices were dissected from E18 mouse embryos in HBSS and treated with 0.25% trypsin-1mM EDTA for 15 min at 37 °C. Tissues were washed, transferred to DMEM seeding medium (DMEM, 10% horse serum,

0.5 mM L-glutamine) and dissociated by 7–8 cycles of aspiration and ejection through a micropipette tip. Neurons were seeded at 250,000 neurons per cm² on coverslips coated with 50 μg/ml poly-D-lysine. After 3 h, the seeding medium was replaced by serum-free neuronal culture medium (Neurobasal medium, 2% B27 supplement, penicillin/streptomycin and 0.5 mM L-glutamine). For Mpc1 downregulation DIV7 neurons were treated with lentiviral particles containing shRNA targeting Mpc1 [ShMpc1 (B)] for a further 7–8 days. Briefly, to prepare viral particles, Hek293T cells were transfected with packaging and envelope expressing plasmids together with PLKO.1-shRNA control (SHC016, Sigma) or targeting Mpc1 with the following sequences: ShMPC1_1: CCGGGCTGCCTTACAAGTATTAAATCTCGA-GATTTAATACTTGTAAGGCAGCTTTTT; ShMPC1_2: CCGGGCTGCCATCAATGATATGAAACTCGAGTTTCATAT-CATT ATGGCAGTTT. After 72 h the culture supernatant was collected, ultracentrifuged at 100,000 × g for 2 h.

## Immunofluorescence

DIV15 primary hippocampal neurons were fixed in 4% paraformaldehyde (PFA) supplemented with 4% sucrose in Dulbecco's phosphate-buffered saline (PBS) solution, permeabilized with Triton 0.1% in PBS, and blocked with 5% bovine serum albumin (BSA). Primary antibody incubation was performed overnight at 4 °C. The specific antibodies used and antibody concentrations are listed below, in the antibody section. After washes with PBS, secondary antibodies (Alexa Fluor, Thermo Fisher) were added, and samples were incubated for 1 h at room temperature. Coverslips were mounted with Fluoroshield (Sigma, cat number F6182). Images were acquired using confocal microscopy. Details of image acquisition have been reported in the section on confocal microscopy below.

## Total protein isolation and purification of the Triton-insoluble fraction

DIV15 primary neurons were homogenized in 0.32 M sucrose, 1 mM NaHCO₃, 1 mM HEPES, 1 mM NaF, 0.1 mM PMSF, and 1 mM MgCl₂ with protease inhibitors (Complete™, GE Healthcare, Mannheim, Germany) and Ser/Thr and Tyr phosphatase inhibitors (PhosSTOP, Roche Diagnostics GmbH). To purify the Triton-insoluble fraction, the lysate was centrifuged at 13,000 × g for 15 min at 4 °C and the pellet was resuspended in 0.5% Triton X-100 and 75 mM KCl and incubated for 15 min at 4 °C. Samples were centrifuged at 100,000 × g for 1 h at 4 °C and the pellet was resuspended in 20 mM HEPES.

## SDS-PAGE and western blot analyses

Protein samples were separated by electrophoresis on acrylamide/bisacrylamide gels at the appropriate concentration, transferred to a nitrocellulose membrane, and probed with the corresponding primary antibodies followed by incubation with appropriate horseradish peroxidase-conjugated secondary antibodies (Bio-Rad). See the list at the end of the section with all the info related to antibodies used. The proteins were detected by chemiluminescence using Clarity Western ECL substrate reagent (Bio-Rad cat number 170-5061) and acquired with a ChemiDoc instrument

(Bio-Rad). Quantification was conducted using the Bio-Rad ImageLab software.

## Oxygen consumption rate analysis

Oxygraphic measurements were performed using a Clark type oxygen electrode (Hansatech, DW1 electrode chamber). For coupled respiration measurements, detached primary neurons were rinsed in pre-warmed PBS (37 °C) and suspended in coupled respiration buffer (2% fatty acids-free BSA, 1 mM sodium pyruvate, and 25 mM D-glucose). Samples were then transferred to the electrode chamber for the oxygen consumption rate measurement. After measuring the basal respiration, the maximal, uncoupled, and maximal uncoupled respiration were evaluated by adding 5 mM ADP, 10 μM oligomycin, and 10 μM CCCP, respectively. All sample values were normalized to total protein content. Mitochondrial respiration was also examined using the Agilent Seahorse XFe24 analyzer and measurements were performed as previously described (Gu et al, 2021). For the Seahorse assays, oligomycin, FCCP, and Rot/AA were used at the concentrations of 2.5 μM, 2.0 μM, and 0.5 μM, respectively.

## Targeted metabolomics and metabolic flux analysis

Metabolomic data were obtained using liquid chromatography coupled to tandem mass spectrometry. We used an API-3500 triple quadrupole mass spectrometer (AB Sciex, Framingham, MA, USA) coupled with an ExionLC™ AC System (AB Sciex). Cells were dysrupted in a tissue lyser for 1 min at maximum speed in 250 μl ice-cold methanol/water/acetonitrile 55:25:20 containing 1 ng/μl [U-$^{13}$C$_6$]-glucose and 1 ng/μl [U-$^{13}$C$_5$]-glutamine as internal standards. Lysates were spun at $15,000 \times g$ for 15 min at 4 °C. Samples were then dried under N$_2$ flow at 40 °C and resuspended in 125 μl ice-cold MeOH/H$_2$O/Acetonitrile (ACN) 55:25:20 for subsequent analyses.

As previously described, quantification of amino acids, their derivatives, and biogenic amines was performed through previous derivatization (Audano et al, 2018). Briefly, 25 μl of each 125 μl sample were collected and dried separately under N$_2$ flow at 40 °C. Dried samples were resuspended in 50 μl phenyl-isothiocyanate, EtOH, pyridine, and water 5%:31.5%:31.5%:31.5%, then incubated for 20 min at RT, dried under N$_2$ flow at 40 °C for 90 min, and finally resuspended in 100 μl 5 mM ammonium acetate in MeOH/H$_2$O 50:50. Quantification of different amino acids was performed using a C18 column (Biocrates, Innsbruck, Austria) maintained at 50 °C. The mobile phases for positive ion mode analysis were phase A: 0.2% formic acid in water and phase B: 0.2% formic acid in acetonitrile. The gradient was T0: 100%A, T5.5: 5%A, T7: 100%A with a flow rate of 500 μl/min. All metabolites analyzed in the described protocols were previously validated by pure standards and internal standards were used to check instrument sensitivity.

Quantification of energy metabolites and cofactors was performed using a cyano-phase LUNA column (50 mm × 4.6 mm, 5 μm; Phenomenex) with a 5.5 min run in negative ion mode with two separated runs. Protocol A: mobile phase A was water and phase B was 2 mM ammonium acetate in MeOH, with a gradient of 10% A and 90% B for all analyses and a flow rate of 500 μl/min. Protocol B: mobile phase A was water and phase B was 2 mM ammonium acetate in MeOH, with a gradient of 50% A and 50% B for all analyses and a flow rate of 500 μl/min.

Acylcarnitine quantification was performed on the same samples using a Varian Pursuit XRs Ultra 2.8 Diphenyl column (Agilent). Samples were analyzed in a 9 min run in positive ion mode. Mobile phases were A: 0.1% formic acid in H$_2$O, B: 0.1% formic acid in MeOH, and the gradient was T0: 35%A, T2.0: 35%A, T5.0: 5%A, T5.5: 5%A, T5.51: 35%A, T9.0: 35%A with a flow rate of 300 μl/min.

MultiQuant™ software (version 3.0.3, AB Sciex) was used for data analysis and peak review of chromatograms. Raw areas were normalized to the areas' sum. Obtained data were then compared to controls and expressed as fold change. Obtained values were considered to be median-scaled metabolite levels. Data processing and analysis were performed using the MetaboAnalyst 5.0 web tool.

For metabolic flux analysis, neurons were treated with 12.5 mM [U-$^{13}$C$_6$]-glucose for 24 h before harvesting. Cells were then treated as described above for metabolite extraction. Quantification of labeled metabolites was performed using a Luna 3 μm HILIC 200 Å, LC Column 150 × 4.6 mm (Phenomenex) at 35 °C on a 7 min run in negative ion mode. Mobile phase A was water and phase B was 2 mM ammonium acetate in MeOH. The gradient was T0: 80%B; T1: 80%B; T1.01: 20%B; T2.0: 20%B; T2.01: 80%B; T7: 80%B with a flow rate of 600 μl/min. Metabolite quantification was performed as described above by increasing the multiple reaction monitoring (MRM) transitions in negative ion mode to 139 to analyze the different isotopomers.

## Untargeted metabolomics

Metabolites from adult hippocampi (neuro-Mpc1-WT or neuro-Mpc1-KO) were extracted as follows: aliquots of 10 mg of tissue were homogenize in 1 mL of cold 70% (v/v) ethanol with a TissueLyser. Extraction was continued with addition of 7 mL of 70% (v/v) ethanol preheated to 75 °C for 2 min, followed by removal of debris by centrifugation (15 min at 4000 rpm at 4 °C). Extracts were stored at −20 °C until mass spectrometry analysis.

Metabolites from $2.10^6$ 17 DIV cortical neurons [SCR (B) or ShMpc1 (B)] were extracted as follows: neurons were washed twice with 2 ml of wash solution (Ammonium Carbonate 75 mM, pH 7.4). Plates were immersed in liquid nitrogen to quench metabolism prior to extraction. Metabolites were extracted by adding 800 ml of extraction solvent (2:2:1 acetonitrile:methanol:water) and incubation at −20 °C for 10 min. Neurons were scrapped on ice and extracts were centrifuged at 13,000 rpm for 2 min at 4 °C. Extracts were stored at −20 °C until mass spectrometric analysis.

Non-targeted metabolomics was performed by flow injection analysis on a 6550 Agilent QTOF instrument as described previously. Briefly, profile spectra were recorded in negative ionization from $m/z$ 50 to 1000 mode at 4 GHz high-resolution mode. Ion annotation was based on accurate masses using a tolerance of 0.001 a.m.u. and KEGG mmu database, accounting systematically for H$^+$ and F$^-$ ions, sodium and potassium adducts, and heavy isotopes.

## Confocal microscopy

Images were acquired with the Zeiss confocal laser scanning microscope LSM900 as a z-stack series, using a ×63 oil immersion

objective (for Sholl analysis and mitochondrial morphology experiments, we used a ×63 oil immersion objective with a zoom of 0.7X) and processed with Fiji software (US National Institutes of Health).

For the analysis spine density and morphology in primary neurons treated with UK5099 alone or in presence of DHPG, images were acquired as z-stack series with a ×60 oil objective using a Nikon A1R confocal microscope.

## Electrophysiology

Whole-cell voltage clamp recordings were obtained with an Axopatch 200-B amplifier and pCLAMP software; mEPSCs were recorded at −70 mV in the presence of 1 mM tetrodotoxin (TTX) at room temperature (20–25 °C) and analyzed using Clampfit software (Pizzamiglio et al, 2016). Currents were filtered at 2 kHz and sampled above 10 kHz. mEPSCs had to exceed a threshold of 8 pA to be included. Multi-unit activity (MU) was detected in cell-attached configuration clamping neurons at −60 mV (Pizzamiglio et al, 2021). External control solutions contained 125 mM NaCl, 5 mM KCl, 1.2 mM $MgSO_4$, 1.2 mM $KH_2PO$, 2 mM $CaCl_2$, 6 mM D-glucose, and 25 mM HEPES/NaOH, pH 7.4. In the case of MU, no TTX was applied in the extracellular solution. Recording pipettes were fabricated from capillary glass using a two-stage puller (Narishige, Japan) and had tip resistances of 3–5 MO when filled with the intracellular solution of the following composition: 130 mM potassium gluconate, 10 mM KCl, 1 mM EGTA, 10 mM HEPES, 2 mM $MgCl_2$, 4 mM MgATP, and 0.3 mM Tris-GTP. Mean Hz is referred to mean frequency.

## SUnSET

Protein synthesis rates were examined using the SUnSET methodology (Schmidt et al, 2009). DIV15 primary neurons were treated for 30 min with 2 μM puromycin; 40 μM anisomycin was used as a control. Cells were washed in PBS and lysed for western blot analysis or fixed in 4% PFA/4% sucrose and stained with puromycin for imaging analysis.

## Synaptoneurosome purification

DIV15 neurons were lysed in 0.32 M sucrose, 1 mM EDTA, 5 mg/ml HEPES pH 7.4, and 2 mg/ml BSA in the presence of protease and phosphatase inhibitors. The lysate was centrifuged for 4 min at $3000 \times g$ at 4 °C. The supernatant was centrifuged at $14,000 \times g$ for 14 min at 4 °C. The pellet was resuspended in Krebs-Ringer buffer (140 mM NaCl, 5 mM glucose, 5 mM KCl, 1 mM EDTA, 10 mM HEPES pH 7.4) and Percoll was added on the top. Samples were centrifuged for 2 min at $14,000 \times g$ at 4 °C. The obtained pellet was then lysed in 100 mM NaCl, 10 mM $MgCl_2$, 10 mM Tris HCl pH 8.0, 1 mM DTT, and 1% Triton X-100 to generate the synapto-neurosome sample.

## RNA extraction and RT-qPCR

RNA was extracted from DIV15 neurons in TRIZOL (Thermo Fisher) according to the supplier's specifications. RT-qPCR was performed with the iTaq Universal SYBR Green One-Step Kit (Bio-Rad, 1725151) using the CFX-384 well plate instrument (Bio-Rad).

The following oligos were used for qPCR analysis:
Rat Histone H3 FOR 5′-CGTTGGAGGAGCTTCGTCTT-3′
Rat Histone H3 REV 5′-GGCCATCTTCTCTCACCCAA-3′
Rat Tubulin a1a FOR 5′-CGCT GTAAGAAGCAACACCTC-3′
Rat Tubulin a1a REV 5′-ACACTCACGCATGGTTGCTG-3′
Rat PSD-95 FOR 5′-ACCAAGATGAAGACACGCCC-3′
Rat PSD-95 REV 5′-ATCACAGGGGGAGAATTGGC-3′
Rat Mpc1 FOR 5′-GACTTCCGGGACTATCTCATG-3′
Rat Mpc1 REV 5′-GTCAGAGAATAGCAACAGAGGG-3′
Rat 36B4 FOR 5′-GGATGACTACCCAAAATGCTTC-3′
Rat 36B4 REV 5′-TGGTGTTCTTGCCCATCAG-3′
Rat Glutamate dehydrogenase FOR 5′-GAGGTCATC-GAAGGCTACCG-3′
Rat Glutamate dehydrogenase REV 5′-ACTCACGT-CAGTGCTGTACC-3′.

## Experimental design and quantification of data

To minimize the possibility of bias in experimental results, randomization and blinding were used in the experimental design, imaging acquisition, and analyses. Cell cultures with viability lower than 80% were excluded from the analyses. Acquisition and quantification of western blotting was performed with computer-assisted imaging (ChemiDoc system and Image lab 4.0 software; Bio-Rad).

Density and morphological analysis of dendritic spines was performed on the total length of the dendrites using ImageJ software to measure spine length, head and neck width (Malinverno et al, 2010). Sholl analysis and the measurement of total dendritic length were performed on neurons transfected with the neurospecific filler mCherry using the Neuroanatomy plug-in of the Fiji freeware software. The Sholl interval was set at 5 μm.

To analyze protein synthesis levels at synapses, we measured the integrated density of puromycin in the dendritic spines, selecting the same area for the ROI for each spine. For each neuron, we analyzed a minimum of 25 spines and averaged these values. Mitochondrial morphology was analyzed as previously described (Audano et al, 2021). Briefly, images of mitochondria were analyzed with Fiji software. Images were smoothed and binarized to analyze the area of single particles. Binaries were then skeletonized, and each skeleton was quantified to obtain the number of networks, junctions, and branches and the length of each particle. Quantification of skeletonized particles was performed with no elimination of loops and endpoints.

## Statistical analysis

Throughout the manuscript, when continuous variables are considered, values are reported as mean ± SE. G Power Software was used to calculate the adequate sample size. The type of parametric or non-parametric test used for each experiment and the corresponding $p$-values, as well the type of adjustment for multiple comparisons (if any), are provided in figure legends. All statistical tests were two-sided and results were assumed to be statistically significant if $p < 0.05$.

Statistical analyses were performed using GraphPad Prism (version 9.3), while for targeted metabolomics data we used the MetaboAnalyst 5.0 webtool to perform an ANOVA followed by Fisher's least significant difference test. We corrected $p$-values with

a Benjamini–Hochberg procedure to control the false discovery rate to adjust for multiple tests.

## Antibodies

The following antibodies were used in this study:
Primary antibodies:

- PSD-95, Abcam ab19257 (WB 1:2000, IF 1:200)
- MAP2, Synaptic System 188004 (IF 1:300)
- β-Actin, Sigma A5441 (WB 1:15,000)
- Puromycin, Millipore MABE343 (WB 1:5000, IF 1:300)
- Histone H3, Proteintech 17168-1AP (WB 1:2000)
- Synaptophysin, Abcam ab32127 (WB 1:50,000)
- Mpc1, Sigma-Aldrich HPA045119 (WB 1:1000)
- Glutamate dehydrogenase Cell Signaling 12793 (WB 1:1000)
- GFAP Cell Signaling 367 (WB 1:3000).

Secondary antibodies:

- Goat anti-rabbit (HRP-conjugated), Bio-Rad 170-6515 (1:10,000)
- Goat anti-mouse (HRP conjugated), Bio-Rad 172-1011 (1:10,000)
- Alexa Fluor 488 goat anti-rabbit, Invitrogen A11034 (1:1000)
- Alexa Fluor 555 goat anti-mouse, Invitrogen A21424 (1:1000)
- Alexa Fluor 647 goat anti-guinea pig, Invitrogen A21450 (1:300).

## Data availability

This study includes no data deposited in external repositories.

## Peer review information

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

## Acknowledgements

We thank Prof. Jean-Claude Martinou and Andres De La Rossa for providing Neuro-Mpc1-WT and -KO mice tissues and for the scientific discussion. We thank Elisa Zianni and Annalisa Longhi for technical assistance, Simona Ligorio, Elisa Bidese, Marlene Luckow and Sara Boccaperta Schiavetti for excellent practical work. This project has received funding from the European Union's Horizon research and innovation program under the MSCA Doctoral Networks 2021, No. 101072759 (FuEl ThE bRaiN In healtThY aging and age-related diseases, ETERNITY), the Italian Ministry of University and Research (PRIN 2017MYJ5TH and PRIN 20202THZAW to MDL, PRIN 2017B9NCSX and PRIN 202039WMFP to EM), from Fondazione Cariplo (grant no. 2018-0511 to EM) and partly from the Fondazione Regionale per la Ricerca Biomedica (FRRB; grant no. CP 20/2018 Care4NeuroRare to NM). This work was also partially supported by the Ministry of University and Research (MUR) Progetto Eccellenza (2022–2027) to the Dipartimento di Scienze Farmacologiche e Biomolecolari "Rodolfo Paoletti", Università degli Studi di Milano and partially by the Italian Ministry of Health with Ricerca Corrente and "5xmille" funds to

NM. S.Pedretti is supported by Fondazione Umberto Veronesi post-doctoral fellowship. S.Pelucchi is supported by PSR2022_DIP_022_AZIONE_A_SPELU. The authors acknowledge the support of DiSFeB Progetto di Eccellenza and of Piano di sostegno alla ricerca (PSR) 2022. FA is economically supported by Italian Ministry of University and Research, PRIN 2017 (5C22WM). Part of this work was carried out at NOLIMITS, an advanced imaging facility established by the Università degli Studi di Milano. The scheme of Fig. 4D and the synopsis image were created with BioRender.com.

## Author contributions

**Laura D'Andrea**: Conceptualization; Formal analysis; Investigation; Writing—original draft; Writing—review and editing. **Matteo Audano**: Conceptualization; Formal analysis; Investigation; Writing—original draft. **Silvia Pedretti**: Formal analysis; Funding acquisition; Investigation. **Silvia Pelucchi**: Formal analysis; Investigation. **Ramona Stringhi**: Formal analysis; Investigation. **Gabriele Imperato**: Formal analysis; Investigation. **Giulia De Cesare**: Formal analysis; Investigation. **Clara Cambria**: Formal analysis; Investigation. **Marine H Laporte**: Resources; Formal analysis; Investigation. **Nicola Zamboni**: Resources; Formal analysis. **Flavia Antonucci**: Formal analysis; Funding acquisition; Writing—original draft; Writing—review and editing. **Monica Di Luca**: Funding acquisition; Writing—review and editing. **Nico Mitro**: Conceptualization; Supervision; Funding acquisition; Writing—original draft; Project administration; Writing—review and editing. **Elena Marcello**: Conceptualization; Supervision; Funding acquisition; Writing—original draft; Project administration; Writing—review and editing.

## Disclosure and competing interests statement

The authors declare no competing interests.

# Expanded View Figures

**Figure EV1.  Gene set enrichment analysis and effect of UK5099 administration to DIV10 hippocampal neurons.**

(A) Gene set enrichment analysis showing that the respiratory chain was the only metabolic pathway among the most enriched gene clusters during neuronal differentiation from neuroblasts (NBs). Key pathways during differentiation are indicated in red. False discovery rate (FDR) < 0.02. (B) Timeline of the experimental procedure. Primary hippocampal neurons were isolated and plated (DIV0) and the transfection with a plasmid to express EGFP was performed on DIV7. UK5099 was added to the medium at DIV10 and dendritic arborization was analyzed at DIV15. (C) Representative confocal images (left panels) of DIV15 primary hippocampal neurons transfected with EGFP and treated at DIV10 with UK5099 (UK5099 DIV10-15) or DMSO as a control (scale bar 30 μm). The graphs on the right show the quantification of the dendritic arborization and total dendritic length ($n = 3$ biological replicates, >20 neurons for each condition; Sholl analysis: multiple t-test, df $= 4$, t $= 4.02$ $*p = 0.0158$, t $= 2.94$ $*p = 0.0423$, t $= 4.05$ $*p = 0.0154$, t $= 4.059$ $*p = 0.0153$, t $= 2.808$ $*p = 0.0483$, t $= 3.361$ $*p = 0.0282$, t $= 3.741$ $*p = 0.0201$, t $= 4.326$ $*p = 0.0123$, t $= 3.891$ $*p = 0.0176$, and t $= 3.589$ $*p = 0.0229$ at 50, 55, 60, 65, 70, 75, 80, 85, 90, and 100 μm from the soma, respectively. Dendritic length: paired t-test t $= 4.838$, df $= 2$, $*p = 0.0402$). (D) Comparison of neuronal arborization (left panel) and dendritic length (right panel) in neurons treated with UK5099 at DIV1 (UK5099 DIV1-15) and DIV10 (UK5099 DIV10-15), expressed as the percentage of the corresponding control conditions. UK5099 administration at DIV1 reduces dendritic length by 32% and the number of intersections at 100 μm by 43% compared to control conditions, while exposure to UK5099 at DIV10 leads to a 25% reduction in dendritic length and a 22% decrease in the number of intersections compared to control cells ($n = 3$ biological replicates; >20 neurons for each condition. Sholl analysis: multiple t-test df $= 6$, t $= 2.602$ $*p = 0.0405$ and t $= 3.222$ $*p = 0.0180$ at 95 and 100 μm from the soma, respectively. Dendritic length: paired t-test t $= 7.937$, df $= 2$, $*p = 0.015$). Data information: Data are reported as mean ± SE.

A

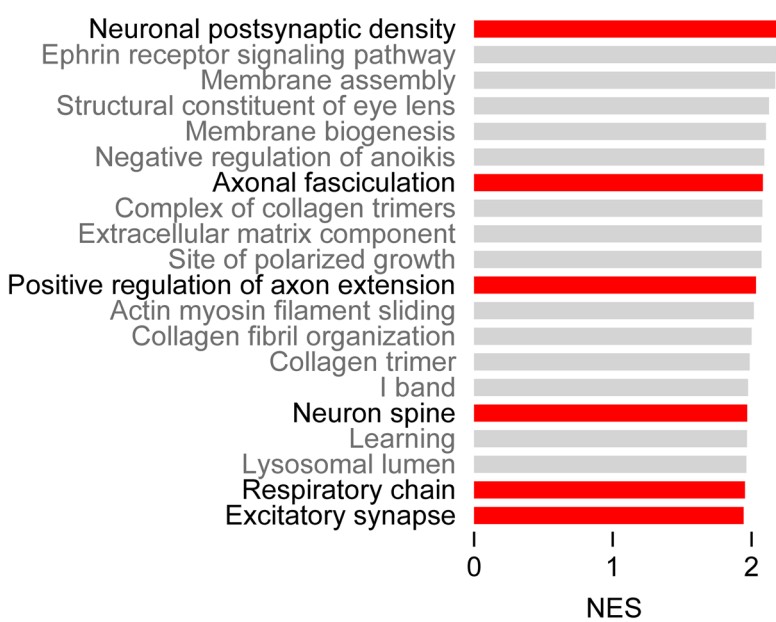

**Neurons vs NBs** (FDR<0.02)

Neuronal postsynaptic density
Ephrin receptor signaling pathway
Membrane assembly
Structural constituent of eye lens
Membrane biogenesis
Negative regulation of anoikis
Axonal fasciculation
Complex of collagen trimers
Extracellular matrix component
Site of polarized growth
Positive regulation of axon extension
Actin myosin filament sliding
Collagen fibril organization
Collagen trimer
I band
Neuron spine
Learning
Lysosomal lumen
Respiratory chain
Excitatory synapse

NES

B

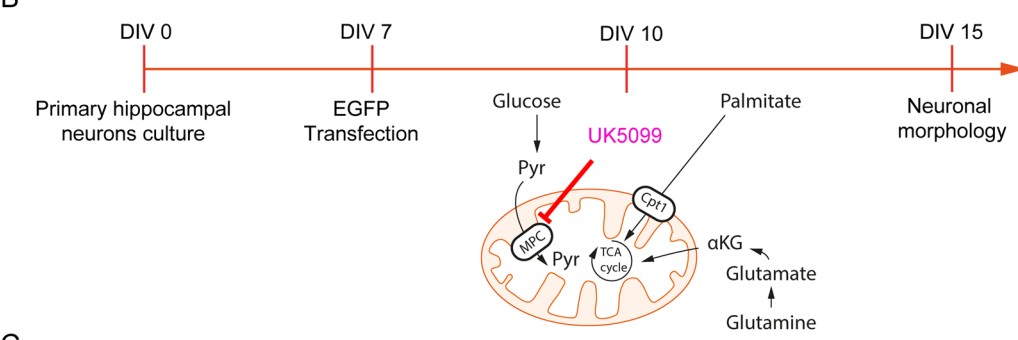

C

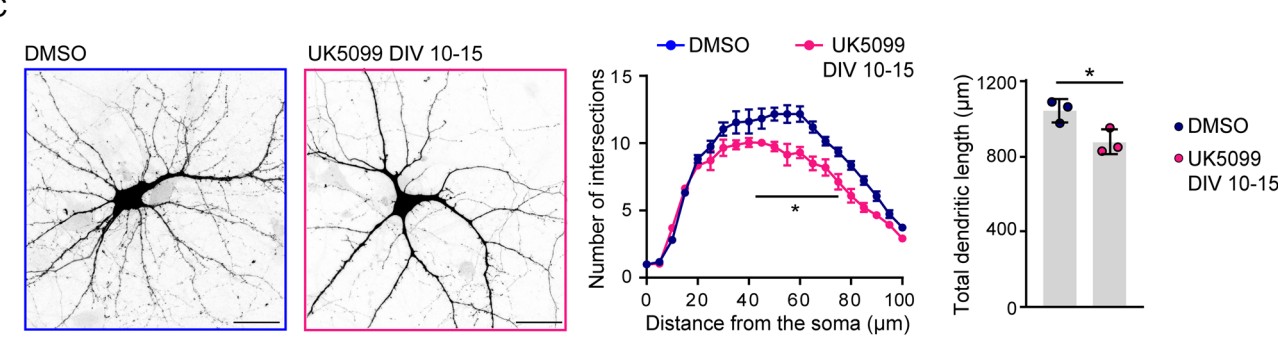

D

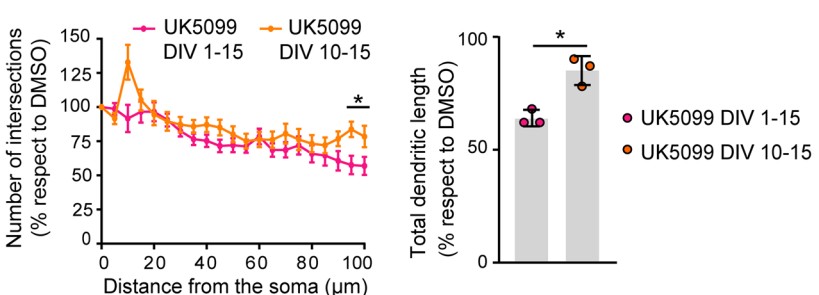

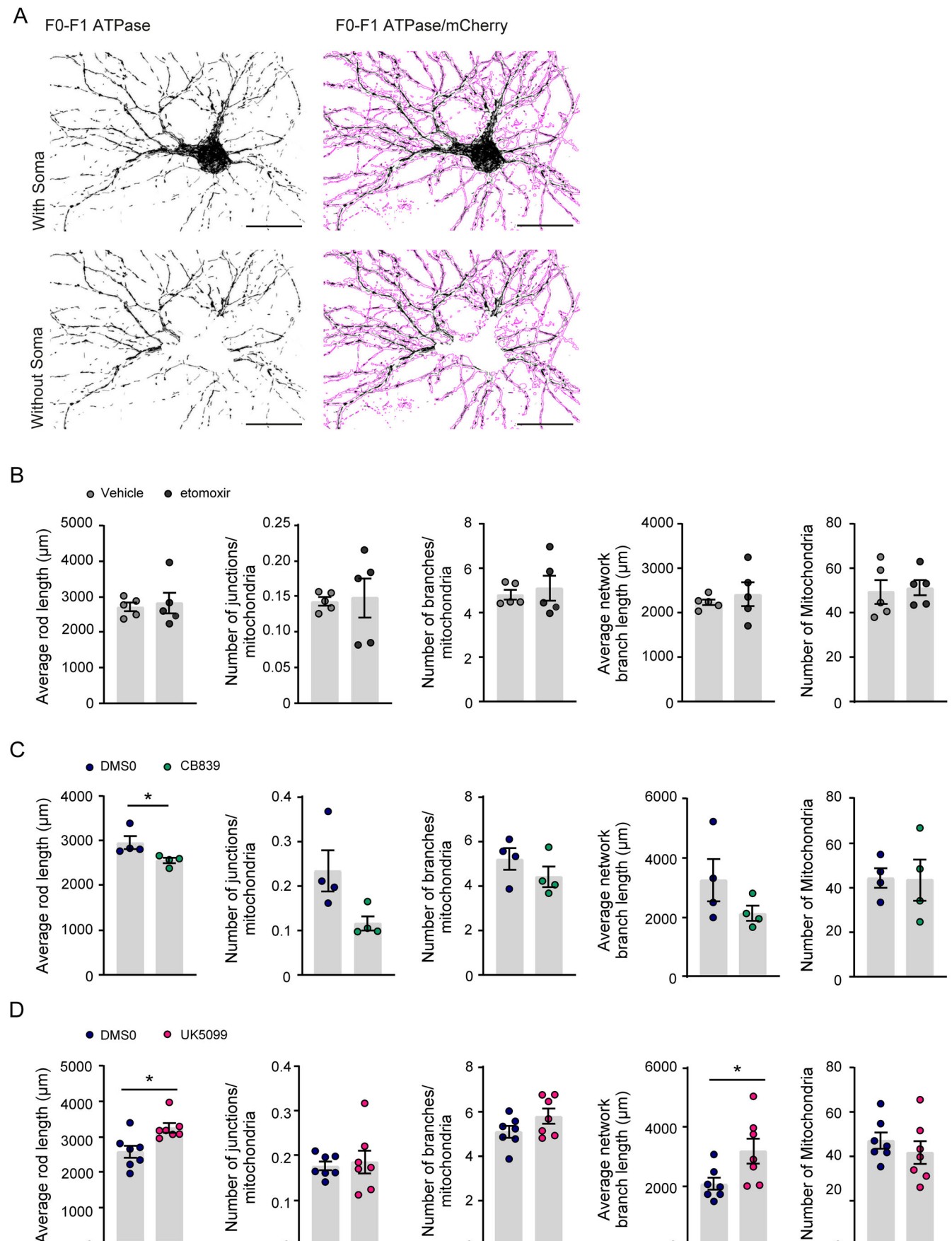

**Figure EV2. Effect of etomoxir, CB839, and UK5099 treatments on mitochondrial morphology.**

(A) Representative confocal images of primary hippocampal neurons transfected with a plasmid to express GFP-tagged subunit 9 of the F0-F1 ATPase protein (to label mitochondria, in black) together with a plasmid encoding the mCherry protein (magenta outline). Analysis was performed along dendrites, excluding the contribution of the soma (scale bar 30 μm). (B–D) Analysis of mitochondrial morphology (average rod length, junction and branch number, and average branch length) in neurons treated with etomoxir (B), CB839 (C), or UK5099 (D) and transfected with the plasmids to express mCherry and GFP-tagged subunit 9 of the F0-F1 ATPase (Etomoxir vs. vehicle: 5 biological replicates, >30 neurons were analyzed for each condition. Paired t-test t = 0.3445 df = 4 $p$ = 0.7479, t = 0.1928 df = 4 $p$ = 0.8565, t = 0.7846 df = 4 $p$ = 0.4766, t = 0.2840 df = 4 $p$ = 0.7905 for mitochondrial average rod length, junction numbers, network branch length and mitochondria number, respectively; Wilcoxon matched signed rank test $p$ = 0.8125 for branch number. CB839 vs. DMSO: 4 biological replicates, >20 neurons were analyzed for each condition. Paired t-test t = 3.389 df = 3 *$p$ = 0.0428 for mitochondrial average rod length, t = 1.205, df = 3 $p$ = 0.3147 for branch number, t = 1.509, df = 3 $p$ = 0.2285 for average network branch length and t = 0.08989 df = 3 $p$ = 0.9340 for mitochondria number; Wilcoxon matched signed rank test $p$ = 0.1250, for junction number. UK5099 vs. DMSO: $n$ = 7 biological replicates, >20 neurons were analyzed for each condition. Wilcoxon matched signed rank test *$p$ = 0.0156 for mitochondrial average rod length, $p$ = 0.9375 for junction number, Paired t-test t = 1.270 df = 6 $p$ = 0.2513, t = 2.669 df = 6 *$p$ = 0.0371, and t = 1.780 df = 6, $p$ = 0.1254 for branch number, average network branch length and mitochondria number, respectively). Data information: Data are reported as mean ± SE.

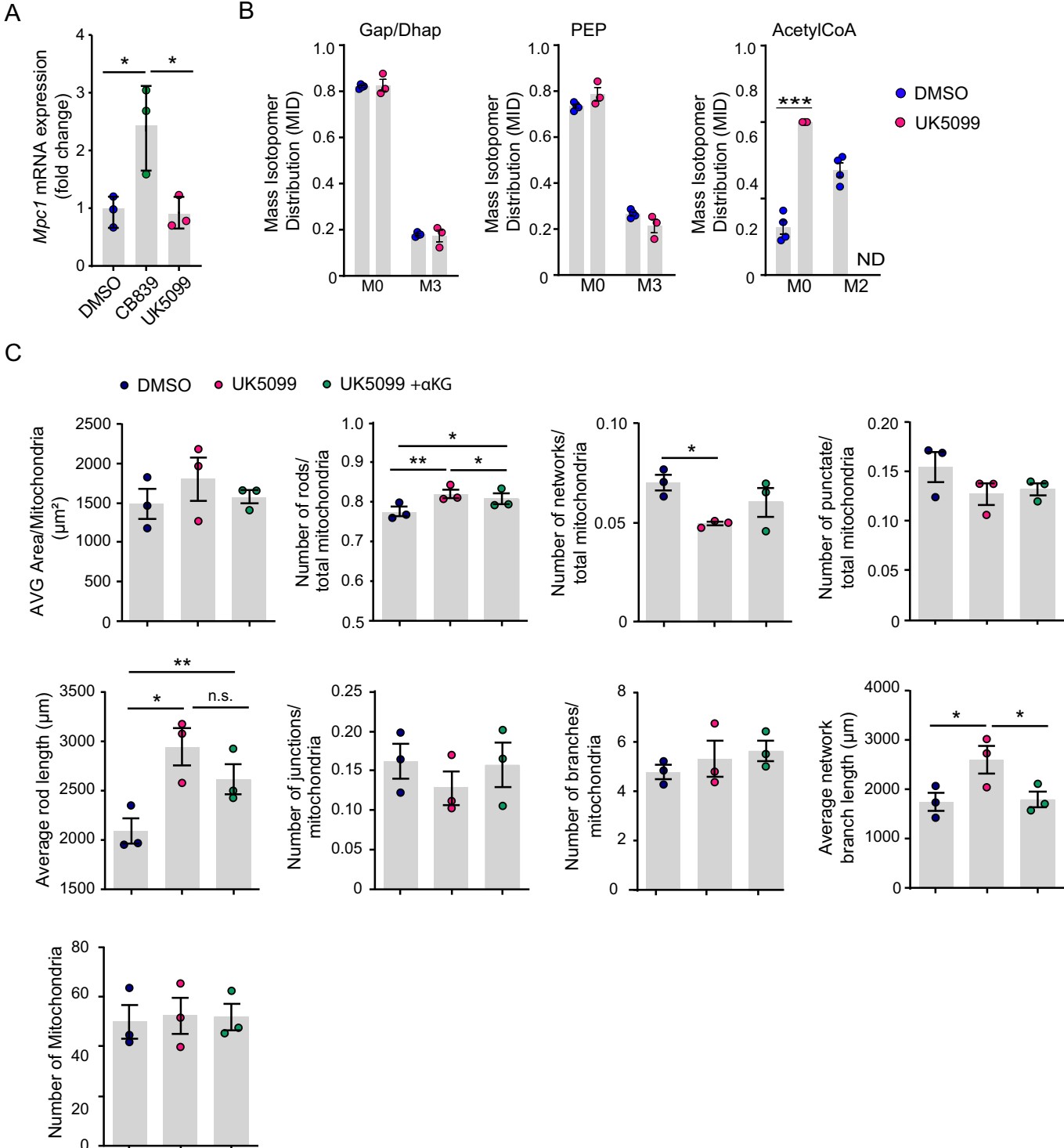

**Figure EV3.** *Mpc1* **expression is upregulated upon CB839 treatment; analysis of isotopomers in UK5099-treated cells and effect of α-ketoglutarate supplementation on mitochondrial morphology in neurons exposed to UK5099.**

(A) RT-qPCR for *Mpc1* mRNA normalized to the housekeeping gene *36B4* in neurons treated with CB839 or UK5099 compared to DMSO. Statistical analysis was performed using one-way ANOVA with Tukey's multiple comparison test ($n = 3$ biological replicates, $*p < 0.05$ vs. DMSO or UK5099). Data are presented as mean ± SD. (B) Mass isotopomer distribution (MID) of glyceraldehyde 3-phosphate/dihydroxyacetone phosphate (Gap/Dhap), phosphoenolpyruvate (PEP), and acetyl-CoA labeled with [U-$^{13}C_6$] glucose. $n = 4$ biological replicates for DMSO and $n = 3$ biological replicates for UK5099. Statistical analysis was performed using Student's t-test. $***p < 0.001$ vs. DMSO. Data are presented as mean ± SD. (C) Mitochondrial morphology analysis along dendrites of neurons treated with DMSO and UK5099 alone or in combination with α-ketoglutarate and transfected with plasmids to express mCherry and the GFP-tagged SV9 F0-F1 ATPase protein. Quantification of the average area of mitochondria, rods, and networks and the number of punctate mitochondria (upper panels, from left to right); mitochondrial average rod length, junction and branch numbers, and average branch length (lower graphs from left to right), number of mitochondria ($n = 3$ biological replicates, ≥20 neurons were analyzed for each condition, Friedman with Dunn's multiple comparison test for average area UK5099 vs. DMSO $p = 0.30748$, UK5099 + αKG vs. DMSO and UK5099 vs. UK5099 + αKG > 0.9999 and for punctate mitochondria UK5099 vs. DMSO $p = 0.123$, UK5099 + αKG vs. DMSO $p = 0.3074$, UK5099 vs. UK5099 + αKG $p > 0.9999$; RM one-way ANOVA with Tukey's multiple comparison test: number of rods UK5099 vs. DMSO $**p < 0.01$; UK5099 + αKG vs. DMSO, UK5099 + αKG vs. UK5099 $*p < 0.05$; number of networks, UK5099 vs. DMSO $*p = 0.0419$, UK5099 + αKG vs. DMSO $p = 0.6943$ and UK5099 + αKG vs. UK5099 $p = 0.4916$; average rod length $*p = 0.047$ UK5099 vs. DMSO, $**p = 0.0056$ UK5099 + αKG vs. DMSO, $p = 0.2215$ UK5099 + αKG vs. UK5099; number of junctions UK5099 vs. DMSO, UK5099 + αKG vs. DMSO, and UK5099 + αKG vs. UK5099: not significant; number of branches UK5099 vs. DMSO, UK5099 + αKG vs. DMSO and UK5099 + αKG vs. UK5099: not significant; average network branch length: UK5099 vs. DMSO $*p = 0.0187$, UK5099 + αKG vs. DMSO $p = 0.4021$, UK5099 + αKG vs. UK5099 $p = 0.0418$; number of mitochondria UK5099 vs. DMSO, UK5099 + αKG vs. DMSO, and UK5099 + αKG vs. UK5099: not significant). Data information: Data are reported as mean ± SE.

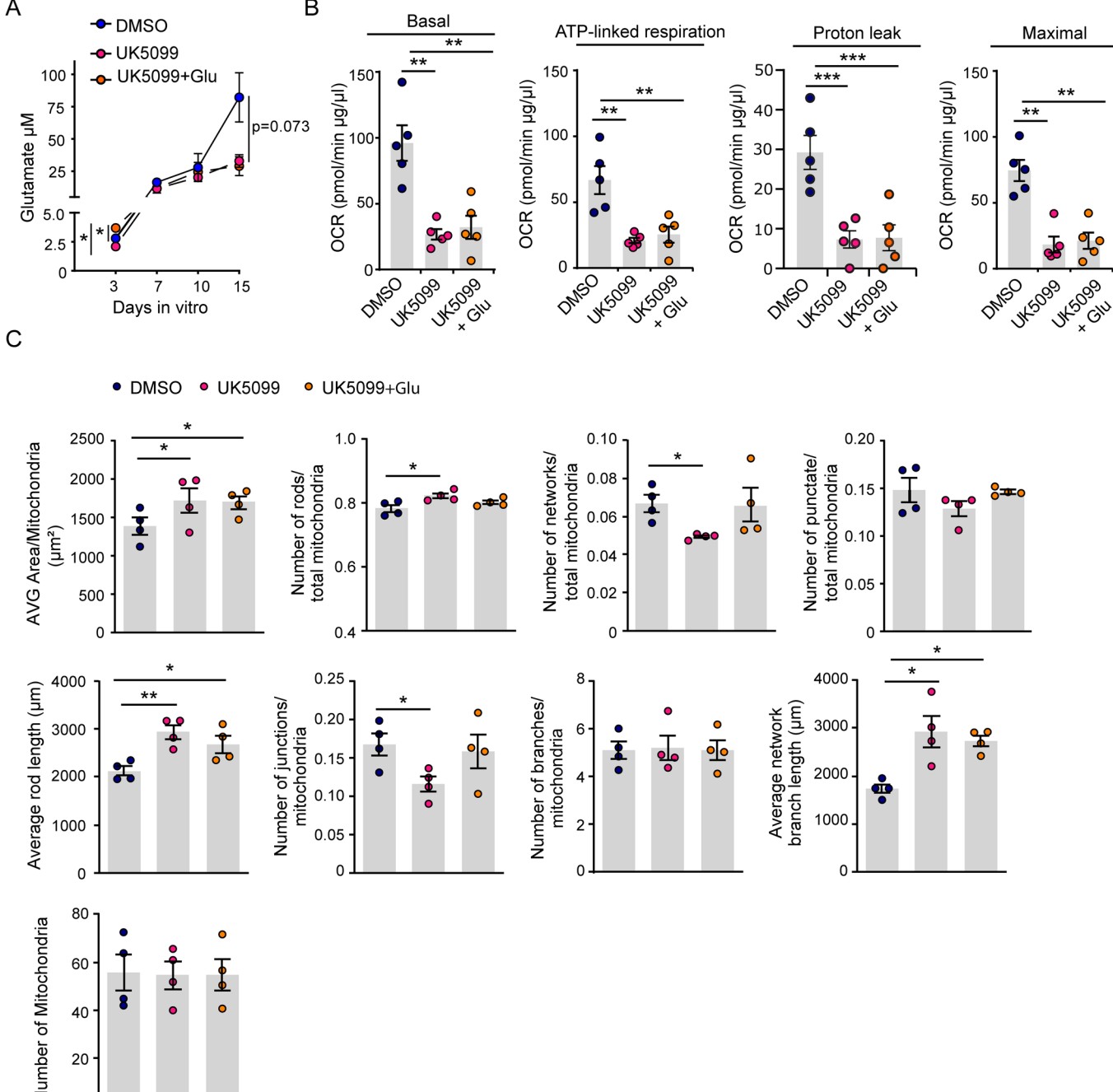

**Figure EV4.   Effect of glutamate supplementation on mitochondrial activity and morphology in neurons exposed to UK5099.**

(A) Glutamate quantification in media from cells treated with DMSO or UK5099 alone or in combination with glutamate at the indicated time points. Statistical analysis was performed using two-way ANOVA with Tukey's multiple comparison test ($n = 3$ biological replicates, $*p < 0.05$ vs. DMSO or vs. UK5099). Data information: Data are presented as mean ± SD. (B) Seahorse XFe24 Cell Mito Stress Test for basal, ATP-linked, proton leak, and maximal uncoupled respiration. Oxygen consumption rate analyses were performed on DIV15 neurons treated with DMSO, UK5099, and UK5099 + glutamate (Glu). ($n = 5$ biological replicates, one-way ANOVA with Tukey's multiple comparison test; Basal: $F_{(2,8)} = 20.47$ df = 8, UK5099 vs. DMSO $**p = 0.0011$, UK5099 + Glu vs. DMSO $**p = 0.0018$, UK5099 + Glu vs. UK5099 $p = 0.8991$; ATP-linked respiration: $F_{(2,8)} = 16.04$ df = 8, UK5099 vs. DMSO $**p = 0.0023$, UK5099 + Glu vs. DMSO $**p = 0.0042$, UK5099 + Glu vs. UK5099 $p = 0.8734$; proton leak: $F_{(2,8)} = 26.14$ df = 8, UK5099 vs. DMSO $***p = 0.0006$, UK5099 + Glu vs. DMSO $***p = 0.0007$, UK5099 + Glu vs. UK5099 $p = 0.9917$; maximal: $F_{(2,8)} = 20.29$ df = 8, UK5099 vs. DMSO $**p = 0.0012$, UK5099 + Glu vs. DMSO $**p = 0.0017$, UK5099 + Glu vs. UK5099 $p = 0.9583$). Data information: Data are presented as mean ± SE. (C) Neurons were treated with DMSO, UK5099, or the combination of UK50999 with Glutamate and transfected with the plasmids to express mCherry and GFP-tagged SV9 F0-F1 ATPase. Quantification of the average mitochondrial area; average number of rod, network, and punctate mitochondria; average rod length; junction, and branch numbers; average network branch length; and mitochondria number was performed (4 biological replicates; >25 neurons were analyzed for each condition. RM ANOVA with Tukey's multiple comparison test for average area, UK5099 vs. DMSO $*p = 0.0350$, UK5099 + Glu vs. DMSO $*p = 0.0172$, UK5099 + Glu vs. UK5099 $p = 0.9502$; number of rods UK5099 vs. DMSO $*p = 0.0251$, UK5099 + Glu vs. DMSO $p = 0.3318$ and UK5099 + Glu vs. UK5099 $p = 0.2942$; number of networks UK5099 vs. DMSO $*p = 0.0495$, UK5099 + Glu vs. DMSO $p = 0.9972$ and UK5099 + Glu vs. UK5099 $p = 0.2798$; average rod length UK5099 vs. DMSO $**p = 0.0027$, UK5099 + Glu vs. DMSO $*p = 0.0308$, UK5099 + Glu vs. UK5099 $p = 0.1285$; number of junctions UK5099 vs. DMSO $*p = 0.0391$, UK5099 + Glu vs. DMSO $p = 0.8747$ and UK5099 + Glu vs. UK5099 $p = 0.1391$; number of branches UK5099 vs. DMSO, UK5099 + Glu vs. DMSO and UK5099 + Glu vs. UK5099 not significant; average network branch length UK5099 vs. DMSO $*p = 0.0498$, UK5099 + Glu vs. DMSO $*p = 0.0152$, UK5099 + Glu vs. UK5099 $p = 0.8614$; mitochondria number UK5099 vs. DMSO, UK5099 + Glu vs. DMSO and UK5099 + Glu vs. UK5099 not significant; Friedmann test with Dunn's multiple comparison test for number of punctate mitochondria UK5099 vs. DMSO $p = 0.4719$, UK5099 + Glu vs. DMSO $p > 0.9999$, UK5099 + Glu vs. UK5099 $p = 0.2313$). Data information: Data are reported as mean ± SE.

                                       

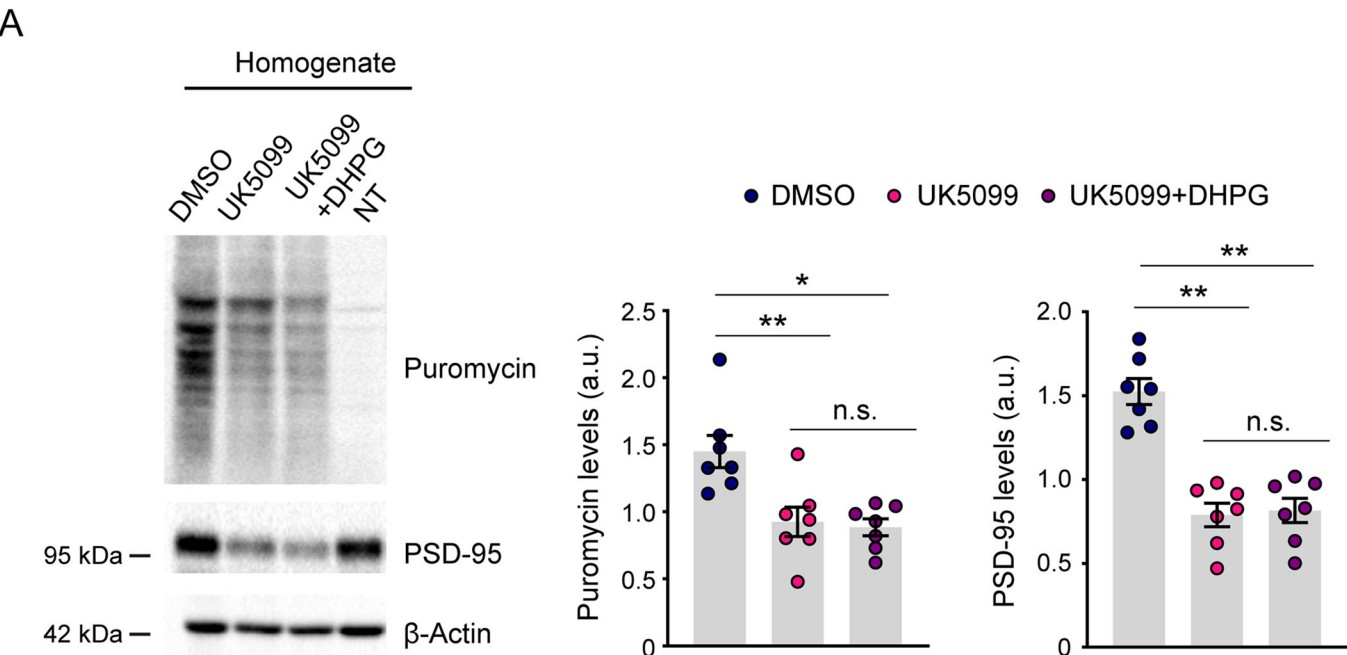

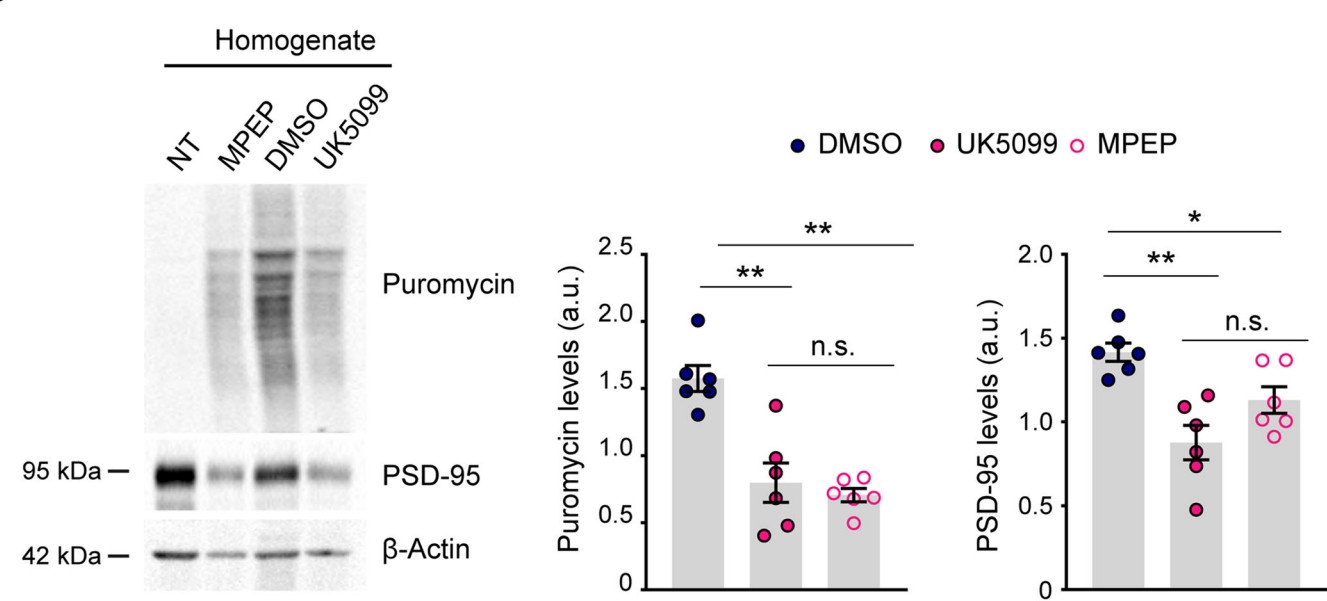

**Figure EV5. Effect of DHPG and MPEP on global protein synthesis and PSD-95 levels in total extract of UK5099-treated neurons.**

(A,B) Representative western blot and quantification of puromycin incorporation and PSD-95 expression in total extract of DIV15 rat primary neurons treated at DIV1 with UK5099 in presece/absence of DHPG (A) and treated with UK5099 and MPEP (B). β-Actin was used as loading control and normalyzer. Data are shown as mean ± SE (Ordinary one-way ANOVA, Tukey's multiple comparison test. A: $n = 7$ biological replicates. Puromycin: UK5099 vs DMSO **$p = 0.0060$, UK5099 + DHPG vs DMSO *$p = 0.0145$, UK5099 + DHPG vs UK5099 $p = 0.9451$; PSD-95: UK5099 vs DMSO **$p = 0.0012$, UK5099 + DHPG vs DMSO **$p = 0.0037$, UK5099 + DHPG vs UK5099 $p = 0.9696$; B: $n = 6$ biological replicates. Puromycin: UK5099 vs DMSO **$p = 0.0043$, MPEP vs DMSO **$p = 0.0017$, UK5099 vs MPEP $p = 0.8467$; PSD-95: UK5099 vs. DMSO **$p = 0.0045$, MPEP vs DMSO *$p = 0.0120$, UK5099 vs MPEP $p = 0.2183$).

