## [Peer Review File · EMBO Reports]

Glucose-derived glutamate drives neuronal terminal differentiation in vitro

Laura D'Andrea, Matteo Audano, Silvia Pedretti, Silvia Pelucchi, Ramona Stringhi, Gabriele Imperato, Giulia De Cesare, Clara Cambria, Marine Laporte, Nicola Zamboni, Flavia Antonucci, Monica DiLuca, Nico Mitro, and Elena Marcello

DOI: [10.15252/embr.202357812](https://doi.org/10.15252/embr.202357812)

Corresponding authors: *Elena Marcello* (elena.marcello@unimi.it) , *Nico Mitro* (nico.mitro@unimi.it)

Review Timeline:

Submission Date:	13th Jul 23
Editorial Decision:	17th Jul 23
Revision Received:	9th Aug 23
Editorial Decision:	26th Sep 23
Revision Received:	1st Dec 23
Accepted:	19th Dec 23

Editor: *Esther Schnapp*

Transaction Report:

Dear Elena,

Thank you for the transfer of your manuscript and your proposed revision plan to EMBO reports.

I have discussed your study and your revision plan with my colleagues here, and we agree that a revised ms - along the lines you suggest - might make a nice contribution to our journal.

I would thus like to invite you to revise your manuscript with the understanding that the referee concerns must be fully addressed and their suggestions taken on board. Please address all referee concerns in a complete point-by-point response. Acceptance of the manuscript will depend on a positive outcome of a second round of review. It is EMBO reports policy to allow a single round of major revision only and acceptance or rejection of the manuscript will therefore depend on the completeness of your responses included in the next, final version of the manuscript.

We realize that it is difficult to revise to a specific deadline. In the interest of protecting the conceptual advance provided by the work, we recommend a revision within 3 months (17th Oct 2023). Please discuss the revision progress ahead of this time with the editor if you require more time to complete the revisions.

- 1) A data availability section providing access to data deposited in public databases is missing. If you have not deposited any data, please add a sentence to the data availability section that explains that.
- 2) Your manuscript contains statistics and error bars based on $n=2$. Please use scatter blots in these cases. No statistics should be calculated if $n=2$.

5) a complete author checklist, which you can download from our author guidelines <https://www.embopress.org/page/journal/14693178/authorguide>. Please insert information in the checklist that is also reflected in the manuscript. The completed author checklist will also be part of the RPF.

6) Please note that all corresponding authors are required to supply an ORCID ID for their name upon submission of a revised manuscript (<https://orcid.org/>). Please find instructions on how to link your ORCID ID to your account in our manuscript tracking system in our Author guidelines <https://www.embopress.org/page/journal/14693178/authorguide#authorshipguidelines>

7) Before submitting your revision, primary datasets produced in this study need to be deposited in an appropriate public database (see <https://www.embopress.org/page/journal/14693178/authorguide#datadeposition>). Please remember to provide a reviewer password if the datasets are not yet public. The accession numbers and database should be listed in a formal "Data

Availability" section placed after Materials & Method (see also <https://www.embopress.org/page/journal/14693178/authorguide#datadeposition>). Please note that the Data Availability Section is restricted to new primary data that are part of this study. * Note - All links should resolve to a page where the data can be accessed. *

12) All Materials and Methods need to be described in the main text. We would encourage you to use 'Structured Methods', our new Materials and Methods format. According to this format, the Materials and Methods section should include a Reagents and Tools Table (listing key reagents, experimental models, software and relevant equipment and including their sources and relevant identifiers) followed by a Methods and Protocols section in which we encourage the authors to describe their methods using a step-by-step protocol format with bullet points, to facilitate the adoption of the methodologies across labs. More information on how to adhere to this format as well as downloadable templates (.doc or .xls) for the Reagents and Tools Table can be found in our author guidelines: <<https://www.embopress.org/page/journal/17444292/authorguide#textformate>>. An example of a Method paper with Structured Methods can be found here: <<https://www.embopress.org/doi/10.15252/msb.20178071>>.

I look forward to seeing a revised form of your manuscript when it is ready.

Referee #1

In this manuscript, D'Andrea et al demonstrate that inhibition of mitochondrial pyruvate transporter impairs required for neuronal dendritic arborization and spine formation. Glutamate supplementation and selective activation of metabotropic glutamate receptors restores these defects. In general, the findings of this manuscript are interesting and abundant of data has been collected and presented. However, all data are collected from an *in vitro* model and some conclusions are not adequately support by the data.

Major comments:

Genetic depletion of Mpc1 in vivo does not significantly affect dendritic development and spine formation in the hippocampus (Petrelli et al., 2023, Science advances). The authors in this manuscript should prove their findings using in vivo approaches rather than in vitro ones only.

We are aware of the importance of proving our findings in *in vivo* model. Indeed, in Fig.6A of the revised version of the manuscript, we reported the metabolomic analysis of hippocampal samples of neuro-Mpc1-KO mice. Our *in vivo* data show that postnatal genetic deletion of Mpc1 in neuronal cells induces the same metabolic effects observed in our *in vitro* model, which are at the core features of our manuscript.

Of note, neuro-Mpc1-KO mice are no longer available for further experiments. Therefore, it is difficult if not impossible for us to generate these animals in our lab in a reasonable time for publication. In fact, the Italian procedure for the use of animal models requires that the researchers submit a detailed protocol (with all the experiments and procedures to be performed) to the Italian Ministry of Health for evaluation. The time needed only for this step is about 8 months. Only after getting approval from the Ministry, we are allowed to generate the neuro-Mpc1-KO animals which means at least another year or more of waiting before performing the requested experiments.

Regarding the recent published paper by Petrelli and colleagues (Petrelli et al., 2023, Science advances), they evaluated the effect of MPC inhibition on adult neural stem progenitor (NSPC) and investigated hippocampal neurogenesis in adult and aged mice. Petrelli and colleagues show that MPC1-deficient NSPCs can give rise to normal mature newborn neurons *in vivo*, because no defects in dendritic arborization and spine density are detected.

The discrepancies with our data are most probably due to the use of a different cellular model.

Our experimental model consists of primary hippocampal neurons isolated at the embryonic state (E18-E19) to study terminal neuronal differentiation and not neurogenesis. Petrelli and colleagues isolated NSPCs from 7-week-old mice, propagated the cells and differentiated them using established protocols. Neural stem cells are biologically different from hippocampal primary neurons because they correspond to different stages of the neurodevelopmental process. For example, it has been shown that hippocampal neural stem cells are more susceptible to neurotoxins than primary

UNIVERSITÀ DEGLI STUDI DI MILANO

DIPARTIMENTO DI SCIENZE FARMACOLOGICHE
E BIOMOLECOLARI "RODOLFO PAOLETTI" - DiSFeB

Direttore: Prof.ssa Monica DiLuca

hippocampal neurons (Pierozan et al., Cell Death and Disease 2020), highlighting diverse responses to stimuli. Therefore, this biological difference in the experimental model could also justify the divergences in the results reported in our manuscript compared to the article by Petrelli and colleagues.

In addition, the results published by Iwata and colleagues (Science 2023) on the effects of MPC inhibition on neuronal arborization and dendritic length are in line with our results. In Fig. 5E, F of the paper by Iwata and colleagues (Iwata et al., Science 2023), the exposure to UK5099 (the MPC inhibitor, that we also used in our work) at DIV3 decreased dendritic growth and complexity in primary mouse neurons, in line with our results.

In light of the relevance of these articles, we discussed these aspects and included specific comments in the manuscript.

Mitochondria were initially small and sparse in newly born mouse neurons and gradually grew in size and quantity during neuronal maturation, accompanying increased mitochondria metabolic activity (Iwata et al., 2023, 2020, Science). However, the authors showed that inhibition of MPC impairs dendritic development but increases mitochondrial mass and network complexity. The authors should address these discrepancies by additional experiments.

We thank the reviewer for her/his comment, and we cited and discussed in our manuscript the data from these two papers (Iwata et al., 2023, 2020, Science).

As reported in the new version of the manuscript, the discrepancies of our results and data provided by Iwata and colleagues can be explained by the use of different cellular models. The articles by Iwata and colleagues investigate mitochondria dynamics during neurogenesis, from cell division to fate commitment (Iwata et al., 2020, 2023, Science). Therefore, they used pluripotent stem cells embryonic stem cell-based system that is a different cellular model compared to our primary hippocampal cultures, that are used to investigate neuronal terminal differentiation. The use of different cellular models can justify discrepancies in the results.

Furthermore, in the articles by Iwata and colleagues, it is not clear whether the analysis of mitochondria was carried out in the soma, in the dendrites or in both compartments. It is well-known that the shape of mitochondria differs among cellular compartments (Nicholas L. Turner, et al. Cell 2022.), therefore we focused our analysis on the mitochondria localized in the dendrites. Mitochondrial fragmentation and fission are related to changes in mitochondrial respiratory capacity and, therefore, we can hypothesize that the changes in mitochondria morphology in dendrites can represent a mechanism adopted to the neurons to compensate MPC inhibition.

Whether MPC inhibition affect the expression levels of glutamate dehydrogenase, which is required for conversion of α -KG into glutamate.

We thank the reviewer for the request of this experiment. We measured the expression levels of glutamate dehydrogenase in neurons upon UK5099 treatment. The results showed that MPC inhibition does not affect the mRNA and protein levels of the glutamate dehydrogenase (Appendix Figure S2).

The a-KG using in this study is not cell-permeable. The authors should use the permeable ones, such as octyl-a-KG.

We apologize with the reviewer for not reporting in the previous version of the manuscript that we used Dimethyl 2-oxoglutarate, a cell permeable form of alpha-ketoglutarate (Betto et al., Nature Genetics 2021). We amended the text and Materials and Methods section accordingly.

The data in Figure 8 do not fully support their claim that glucose-pyruvate-glutamate axis sustains local protein synthesis at synapses, because the inhibition of MPC function impairs global protein synthesis.

We agree with the reviewer that the link between the glucose-pyruvate-glutamate axis and the regulation of protein translation needs further elucidations and therefore we performed additional experiments.

Our data show that blunting the activity of MPC affects global protein synthesis and suggest that the translation of the synaptic protein PSD-95 is locally reduced. The supplementation of glutamate induces a full rescue of PSD-95 levels in synaptoneurosomes (current Figure 7F), but not in total homogenate (current Figure 7A), indicating that glutamate acts locally in the control of PSD-95 expression. However, the effect of glutamate supplementation was partial when we analyzed the results of SUnSET experiments in the synaptoneurosomes purified from neurons exposed to UK5099 (current Fig. 7D).

Therefore, considering that glutamate can activate different classes of receptors and, thereby, can trigger multiple biological pathways, we focused on group I metabotropic Glutamate Receptors (mGluRs), because their activation controls local protein synthesis in neurons (Lüscher & Huber, Neuron 2010; Costa-Mattioli, Neuron 2009).

Consistent with the results of glutamate-mediated rescue experiments, we observed that treating neurons exposed to UK5099 with DHPG, a synthetic agonist of group I mGluRs, did not affect the reduction of the overall translational rate (Fig. EV5A). However, remarkably, DHPG treatment fully restored local protein translation and PSD-95 protein levels to normal specifically at synapses (Fig. 8B). Furthermore, the activation of group I mGluRs signaling fully recovered the impairments of neuronal maturation (Figure 8C), thus indicating that the activation of this class of glutamate receptors is critical to restore the defects due to the reduced generation of glucose-derived glutamate.

"Pyruvate transport into mitochondria impairs glutamate synthesis" in row 198 is ambiguity. This manuscript shows that MPC inhibition decrease pyruvate transport into mitochondria caused reduction of glutamate.

We agree with the reviewer, and we rephrased as follow: MPC inhibition leads to decreased pyruvate transport into mitochondria that is associated with impaired glutamate synthesis.

UNIVERSITÀ DEGLI STUDI DI MILANO

DIPARTIMENTO DI SCIENZE FARMACOLOGICHE
E BIOMOLECOLARI "RODOLFO PAOLETTI" - DiSFeB

Direttore: Prof.ssa Monica DiLuca

Minor concerns:

The images of mitochondrial morphology in figure 3 and EV3 need highly zoom-in images.

We now provide images at higher magnification.

The images in figure 7 and 8 showed the entire single neuronal morphology would be better to exhibit the differences.

We now provide other images to better show the differences.

PSD-95 protein levels show high variation in western blot results. For example, two DMSO treatment groups represent a huge difference in figure 2B.

We agree with the reviewer, and we performed additional treatments to confirm our results.

MPC inhibition decrease spine density and impair mEPCS and action potential. Whether the spine shape is also affected?

We appreciate the reviewer's suggestion. Therefore, we included in the revised version of the manuscript the analysis of spine morphology in neurons exposed to UK5099 and the effect of glutamate and DHPG administration (see Fig. 8D).

Referee #2

Andrea et al. explore how metabolic pathways affect neuronal differentiation, focusing on the impact of mitochondrial transport of pyruvate on hippocampal neuron maturation in vivo. The authors first observe that the pharmacological inhibition of mitochondrial pyruvate carrier (MPC) results in decreased neurite growth/complexity and PSD95-positive puncta, while inhibition of carnitine palmitoyltransferase or Glutaminase Glis1 were without effect. They next find that Glis-1 inhibition or MPC inhibition induces a decreased mitochondria activity (OCR) and morphological changes in mitochondria. Using metabolic measurements they find that MPC inhibition leads to impaired alpha-ketoglutarate and glutamate production. The MPC inhibitor-induced neuronal development deficits are partially rescued by glutamate supplementation but not by alpha-ketoglutarate. Similar effects on glutamate and neuronal development are found following genetic or KD of MPC. Overall these data indicate that MPC is required for normal neuronal development, which is in part linked to defective production of Glutamate. From these results, the authors conclude that neuronal differentiation is regulated by glutamate production through glucose-pyruvate-glutamate axis. They then explore the underlying mechanisms, focusing on protein translation. They find that MPC inhibition leads to strong reduction of global and synaptic translation rates, which are very partially rescued by glutamate. Finally they test whether the glutamate

UNIVERSITÀ DEGLI STUDI DI MILANO

DIPARTIMENTO DI SCIENZE FARMACOLOGICHE
E BIOMOLECOLARI "RODOLFO PAOLETTI" - DiSFeB

Direttore: Prof.ssa Monica DiLuca

defect is linked to its role as neurotransmitter, and find that mGluR1 agonists can partially rescue the impact of MPC1 on neuronal morphogenesis.

Overall this is an interesting manuscript on an important topic. Overall the experiments are original, well-thought and well executed. However the data do not support all the conclusions put forward by the authors. More experimental work is needed to solidify the conclusions, some of which could be toned down in abstract and discussion.

Major points are explained below.

1. The functional evidence that inhibition of MPC affects many aspects of neuronal is quite convincing but it is all based on *in vitro* experiments. As metabolism is quite sensitive to culture conditions and does not always mimic the *in vivo* context this is a serious limitation. While it is appreciated that not all experiments could be performed *in vivo*, at least the impact of MPC inhibition could be performed using the mouse genetic model or KD approaches. The ability to perform functional experiments *in vivo* would greatly strengthen the key conclusions here. If this constitutes too much work then the authors should at least discuss carefully this aspect in the discussion and put 'in vitro' in the title.

We agree with the reviewer, and we are aware of the importance of proving our findings in *in vivo* model. Indeed, In Fig.6 of our manuscript, we reported the impact of Mpc1 genetic deletion on metabolism *in vivo*. The metabolomic analysis of hippocampal samples of neuro-Mpc1-KO mice (De La Rossa A. et al. Elife 2022) show that postnatal genetic deletion of Mpc1 in neuronal cells induces the same metabolic effects observed in our *in vitro* model. Therefore, the core feature of our manuscript and the main metabolism results are confirmed *in vivo*, strengthening the conclusions.

Of note, neuro- Mpc1-KO mice are no longer available for further experiments. Therefore, it is difficult if not impossible for us to generate these animals in our lab in a reasonable time for publication. In fact, the Italian procedure for the use of animal models requires that the researchers submit a detailed protocol (with all the experiments and procedures to be performed) to the Italian Ministry of Health for evaluation. The time needed only for this step is about 8 months. Only after getting approval from the Ministry, we are allowed to generate the neuro-Mpc1-KO animals which means at least another year or more of waiting before performing the requested experiments.

According to the reviewer suggestion, we discussed the limitations of our work.

2. As metabolism can have pleiotropic effects during development, the authors should provide more data on the impact of the chemical compounds used on neuronal number/density and proportion of cell types (progenitor vs neuron vs glial cells). This is key to exclude other effects than those on neuronal maturation, and also because the purity of cell preparation is crucial to interpret correctly OCR and metabolic measurements. The authors should provide the relevant information on number and proportion of cells/neurons for these experiments.

We thank the reviewer for his/her suggestion and performed additional experiments to address this issue. We performed Western Blot analyses on lysates of cell cultures at DIV15 using specific markers: neuronal nuclear protein (NeuN) and Glial Fibrillary Acidic Protein (GFAP) to detect neurons and astrocytes respectively. The treatment with etomoxir has no effect on neurons and astrocytes present in our cultures (Appendix Figure S1). The inhibition of Gls-1 increases the percentage of astrocytes, while blunting MPC activity has a dramatic effect on both neurons and astrocytes, since UK5099 decreases the percentage of both cell populations in the culture (Appendix Figure S1). Considering that MPC blocking does not change the ratio between the neurons and astrocytes, the interpretation of metabolic effects induced by UK5099 treatment is not affected by modifications in the proportion of cell types.

3. Dendritic spine defects that are shown at the end of the manuscript should be quantified in full - a single picture is clearly not enough.

We now provide the uncropped images of the dendrites, as requested by the reviewer and we have analyzed the spine morphology defects induced by UK5099 treatment (Fig. 8D).

4. Although the majority of experiments used MPC inhibitor, the author claimed that "glucose"-derived glutamate regulates neuronal differentiation. However Lactate is another source of pyruvate that should be mentioned and discussed.

We thank the reviewer for her/his suggestion. In the revised version of the manuscript, we discussed the key role of lactate as another fuel source.

5. In Fig 8G, the authors reported that MPC inhibitor-induced deficits in dendrite development were rescued by mGluRs agonist DHPG treatment. But one crucial missing piece of data is whether DHPG treatment has the expected impact (rescue) on protein translation during neuronal development? The authors should assess the impact of DHPG treatment on protein translation in MPC inhibitor-treated neurons. It would also informative to know whether DHPG treatment also rescued synapse development impairment in MPC inhibitor treated cells.

According to reviewer request, we evaluated the effect of mGluRs activation to rescue protein synthesis defects and spine defects dependent on MPC inhibition.

We analysed global and local protein synthesis rate by SUnSET experiments in neurons exposed to UK5099 in the presence of the mGluR I selective agonist, DHPG. We observed that treating neurons exposed to UK5099 with DHPG did not have any effect on the overall translational rate reduction (Fig. EV5A). However, remarkably, DHPG treatment fully restored local translation and PSD-95 protein levels to normal specifically at synapses (Fig. 8B).

Furthermore, we carry out the analysis of spine morphology in neurons exposed to UK5099 and the effect of DHPG administration, that was capable of completely rescuing spine maturation (Fig. 8D).

6. The effects of Glutamate on translation rescue are overinterpreted. In Figure 8D there is no rescue of translation, contrary to what is written in the text. The increase of PSD95 protein levels, (8F) could be related to synapse effects distinct from translation. And the rescue shown in 8D is not overwhelmingly convincing, given the the difficulty of this technique to measure accurately protein translation on cellular samples. It seems much more adequate for the authors to acknowledge that the exact mechanisms by which glutamate impacts on neuronal development remain to be fully explored, and likely include in part protein translation, but many other possibilities should be investigated in more depth. In absence of additional data on this mechanistic question, the authors should discuss their data accordingly, by presenting the data in a much more careful way in abstract, summary at the end of intro, results and discussion.

We agree with the reviewer that the effect of glutamate on translation is an issue to strengthen.

To this aim we have performed additional experiments focusing on the activation of a specific class of glutamate receptors. Indeed, the supplement of glutamate to UK5099-treated neurons can activate different classes of glutamate receptors and, thereby, can trigger different signaling pathways. Considering that protein synthesis control in neurons is strictly dependent on the activation of group I mGluRs (Lüscher & Huber, Neuron 2010; Costa-Mattioli, Neuron 2009), we performed rescue experiments exposing UK5099-treated cells to the mGluR agonist DHPG.

In UK5099-exposed neurons, DHPG restores local protein translation and PSD-95 protein levels in synaptoneurosomes (Fig. 8B), without affecting the rate of global protein translation (Fig. EV5A).

In addition, we tested the effects of MPEP (mGluRs antagonist) on protein translation. We found that blocking mGluRs recapitulates UK5099 effect on protein synthesis and on PSD-95 protein levels reduction.

These results support the hypothesis that glutamate, through the activation of mGluR, can promote the local protein translation mechanisms and, thereby, rescues the defects in neuronal morphogenesis and synaptogenesis caused by MPC blocking.

7. Finally, can the authors block the rescue effects of Glutamate by using mgluR antagonists? This would be important to support the claim that this the effect on glutamate receptors is the main one here, instead of parallel effects not directly linked to metabolic regulation of glutamate. Either way would be interesting, but if not, then again the authors should present and discuss their data in a more careful way leaving open other interpretations and possibilities to explain their data.

To address the issue raised by the reviewer, we attempted to use MPEP (mGluR antagonist) in combination with glutamate in UK5099-treated cells; however, neuron viability was dramatically affected and, thereby, no analysis can be carried out in these conditions (please see figure below).

Figure 1.

Effect of MPEP administration on global protein synthesis and PSD-95 expression in neurons exposed to UK5099 supplemented with glutamate.

Representative western blot and quantification of puromycin incorporation and PSD-95 expression in total extract of DIV15 rat primary neurons treated at DIV1 with UK5099 and UK5099 supplemented with Glutamate in presence/absence of MPEP. β-Actin was used as loading control. Data are shown as mean ±SE (one way ANOVA, Tukey's multiple comparison test, n= 5 independent experiments Puromycin: DMSO vs UK5099 p=0.0188 , DMSO vs UK5099+Glu p=0.2100, DMSO vs UK5099+Glu + MPEP p=0.0386; UK5099 vs UK5099 +Glu p=0.9885; UK5099 vs UK5099+ Glu + MPEP p=0.5982 , UK5099+ Glu vs UK5099 + Glu +MPEP p=0.0175 PSD-95: DMSO vs UK5099 p=0.0221 , DMSO vs UK5099+Glu p=0,0099, DMSO vs UK5099+Glu + MPEP p=0,0190; UK5099 vs UK5099 +Glu p=0.4438; UK5099 vs UK5099+ Glu + MPEP p=0.1246 , UK5099+ Glu vs UK5099 + Glu +MPEP p=0.1081.

Minor comments:

-Fig 1B,C,D, 2A, 3C,D,E, 5D and 6D: Scale bar is missing from figure.

We added the scale bar, as requested to all the panels of the imaging figures.

-Fig 3E: Position of asterisk and bar is not optimal.

We corrected the asterisk and bar position.

-Fig EV2B,C: "UK5099 DIV1" and "UK5099 DIV10" is misleading. It would be better to describe it in another way. For example: UK5099 DIV1-15, UK5099 DIV10-15 etc.

UNIVERSITÀ DEGLI STUDI DI MILANO

DIPARTIMENTO DI SCIENZE FARMACOLOGICHE
E BIOMOLECOLARI "RODOLFO PAOLETTI" - DiSFeB

Direttore: Prof.ssa Monica DiLuca

We amended the names of the experimental conditions according to reviewer's suggestions.

- The impact of MPC inhibition on neuronal development is consistent with a recent paper on cortical neurons (Iwata et al. Science 2023), and related to a paper on adult hippocampal neurogenesis (Petrelli Science Advances 2023), which should be cited and discussed accordingly.

We thank the reviewer for his/her comment. We discussed the two papers (Iwata et al. Science 2023 and Petrelli Science Advances 2023) in the revised manuscript.

Referee #3

Cellular metabolism is emerging as a central driver of cellular differentiation and maturation. This manuscript describes how pyruvate drives affects maturation parameters of hippocampal neurons. The starting point of this study is a comparison of the requirement of different fuels for the TCA cycle, i.e. fatty acids, pyruvate, and glutamine for hippocampal neuron dendrite and spine development. To this end the authors pharmacologically inhibit mitochondrial fatty acid import, mitochondrial pyruvate import, and inhibition of mitochondrial glutaminase usage. The authors observe that inhibition of pyruvate import has the strongest effect on these parameters while inhibition of FA import has little impact on these parameters. The authors also investigate mitochondria morphology parameters and again find that in particular blockage of pyruvate import has a strong effect on mitochondria morphology in hippocampal neurons. The authors then continue to investigate using measurement of oxygen consumption assays, targeted metabolomics and incorporation of C13 labeled Glucose how the different metabolites are utilized. They describe that Pyruvate is in particular required for the generation of glutamate. Notably, they also find that supplementing glutamate ameliorates the pyruvate import block-induced dendritic and spine phenotype. Moreover the authors describe that the genetic inhibition of the MPC1, i.e. the transporter for mitochondrial pyruvate import, results in a comparable phenotype. Finally, the authors describe that the inhibition of pyruvate driven glutamate synthesis results in decreased translation and that stimulation of glutamate receptors rescues the dendrite and spine phenotype of pharmacological inhibition of pyruvate import.

This manuscript is well conceived and provides very interesting and significant insight into the regulation of neuronal maturation / development by specific metabolic pathways. The study is largely convincing.

There are, however, a few major concerns that the authors should address:

1. The authors have to provide the information on how / at which concentration the compounds were dissolved in DMSO. This is in particular important, as the authors appear to use only a single control group (DMSO treated) for the UK5099 and CB 839). The authors have to describe the amount of DMSO that was added to the cultures. The single DMSO control is only valid if the same final concentration of DMSO was added to the cultures.

We thank the reviewer for this comment. DMSO was used at a final concentration of 0.1%, CB839 and UK5099 were diluted accordingly.

2. Statistics: The authors perform independent experiments and then use the single individual data points to calculate statistics. Firstly: it is unclear, how many data points stem from which experiment. Secondly and more importantly, the way the statistics is performed, inflates the number of data points. I suggest that statistics based on the averages from the individual experiments is more appropriate. It is also unclear why the authors have in some cases (e.g. Figure 2 b and c) uneven numbers of independent experiments. Ideally the number of independent experiments should be matched.

Regarding the statistical issues raised by the reviewer, we used statistical analysis commonly used in the literature. It is common practice to select several neurons from a few cultures to perform imaging analysis. For instance, in the article by Iwata and colleagues (Science, 2020), the authors reported quantified mitochondria length showing in the graph data point representing an individual cell average mitochondrial size from 3 biological replicate experiments (Figure 1C, 1E, 1G, 2C, 3A, 4E, 4G, 4I). In the article by Petrelli and colleagues (Science Advances, 2023), the authors reported the analysis of dendritic length (Fig. 5I) and spine density (5J) showing data points corresponding to the single neurons (n=20-30) and indicated the number of mice per group (n=4) in the legend.

Therefore, to clarify data analysis in our manuscript, in the legend we reported the number of cells analyzed for each condition and the information regarding from the number of cell cultures they stem from.

Regarding Fig. 2B and 2C, we agree with the reviewer, and we performed additional experiments to provide more robust results with reduced variability.

3. The authors state in the introduction that"... Glucose is the main fuel substrate used by neurons...." There is a large body of evidence that lactate may be the main fuel, this has to be mentioned and discussed.

In the revised version of the manuscript, we discussed the key role of lactate as another fuel source.

4. The authors propose that glutamate derived from glucose/pyruvate signals via group I mGluRs to drive neuronal differentiation. To further substantiate that glutamate - group I mGluRs signaling drives neuronal differentiation, the authors

UNIVERSITÀ DEGLI STUDI DI MILANO

DIPARTIMENTO DI SCIENZE FARMACOLOGICHE
E BIOMOLECOLARI "RODOLFO PAOLETTI" - DiSFeB

Direttore: Prof.ssa Monica DiLuca

should inhibit mGluR signaling and determine whether this phenocopies inhibition of pyruvate import.

To address the issue raised we took advantage of an antagonist of mGluR (MPEP). We have tested the effect of MPEP on protein translation and we found that blocking mGluRs recapitulates UK5099 effect on protein synthesis and on PSD-95 levels (Fig. EV5B).

5. The [U-¹³C₆]-glucose experiments and the results are difficult to follow for a non-biochemist. As this manuscript is of considerable interest to neuroscientists I would like to ask the authors to better illustrate the pathways through which the mentioned metabolites are generated. Figure 5A and B are poorly described and it is difficult to understand what the TCA scheme describes. For better understanding, why the M2 vs M3 analysis is important in particular in the context of the failure of beta-oxidation to supply glutamate, it would be beneficial to illustrate for the non-biochemist where pyruvate vs. Fatty acids are able to enter the synthesis of glutamate.

We have generated a more detailed diagram (Fig. 4D) illustrating the contribution of glucose-derived carbons to the synthesis of alpha-ketoglutarate and glutamate. This enhanced diagram is aimed at providing a clearer explanation and better conveying the findings of our [U-¹³C₆]-glucose tracing experiments.

In Fig. 4A, we tested the impact of fatty acid beta oxidation inhibition on neuronal metabolism and maturation by means of the carnitine palmitoyl transferase blocker, the rate limiting step in fatty acid beta oxidation, such as Etomoxir (Fig. 4A). In this context, glutamate levels were not affected nor neuronal terminal differentiation (Fig. 4A, 1 and 2). On this ground, it is important to note that, to the best of our knowledge, we are unable to capture the request of this reviewer to demonstrate the respective contributions of pyruvate vs. fatty acids to the synthesis of glutamate in the context of MPC inhibition.

Minor comments:

a) The authors also state that "...a detailed investigation on the roles of the main metabolic pathways during hippocampal neuronal differentiation is warranted....", because "...only one report has focused on cortical neuron differentiation....". I would remove this part of the justification for the study. The study by itself does provide important new insight into the role of different metabolites and metabolic pathways in neuronal differentiation. This convoluted justification is not needed.

We removed this sentence according to the reviewer request.

Dear Prof. Marcello,

Thank you for the submission of your revised manuscript. We have now received the enclosed reports from the referees. As you will see, both referees 2 and 3 still have a few more suggestions that I would like you to address and incorporate before we can proceed with the official acceptance of your manuscript. Please provide a point-by-point response to all comments below with your final ms.

A few editorial requests will also need to be addressed:

- Please correct the COI subheading to "Disclosure and Competing Interest Statement"
- Please remove the author credits from the ms file. All author credits should be entered into our online ms handling system.
- Please add all funding info when you upload the final ms. Some info is currently missing in our online system.
- Please add a table of content with page numbers to the Appendix file.
- Please remove the EV tables from the ms and upload them as individual files called EV Table 1, etc.
- I attach to this email a related ms file with comments by our editors. Please address all comments in the final ms.
- I would like to suggest some minor changes to the abstract. Please let me know whether you agree with the following:

Neuronal maturation is the phase during which neurons acquire their final characteristics in terms of morphology, electrical activity, and metabolism. However, little is known about the metabolic pathways governing neuronal maturation. Here, we investigate the contribution of the main metabolic pathways, namely glucose, glutamine, and fatty acid oxidation, during the maturation of primary rat hippocampal neurons. Blunting glucose oxidation through genetic and chemical inhibition of the mitochondrial pyruvate transporter reveals that this protein is critical for the production of glutamate, which is required for neuronal arborization, proper dendritic elongation, and spine formation. Glutamate supplementation in the early phase of differentiation restores morphological defects and synaptic function by rescuing synaptic local translation. [Is causality shown in the ms?] Furthermore, the selective activation of metabotropic glutamate receptors fully restores the impairment of neuronal differentiation due to the reduced generation of glucose-derived glutamate. Fatty acid oxidation does not affect neuronal maturation. Whereas glutamine metabolism is important for mitochondria, it is not for endogenous glutamate production. Our results provide insights into the role of glucose-derived glutamate as a key player in neuronal terminal differentiation.

EMBO press papers are accompanied online by A) a short (1-2 sentences) summary of the findings and their significance, B) 2-3 bullet points highlighting key results and C) a synopsis image that is exactly 550 pixels wide and 200-600 pixels high (the height is variable). You can either show a model or key data in the synopsis image. Please note that text needs to be readable at the final size. Please send us this information along with the final manuscript.

Kind regards,
Esther

Referee #1:

The authors have performed additional experiments for the revision. I believe all of my questions have been addressed quite well.

Referee #2:

The authors have addressed many of the points raised in the first round of review, but some important points remain

unaddressed, as explained below.

1. It is appreciated that it will be difficult for the authors to obtain in vivo evidence. However in this case the fact that all data obtained are in vitro data should be specified in the title or abstract, as requested before.
2. As requested before, given that metabolism can have pleiotropic effects during development, the authors should provide more data on the impact of the chemical compounds used on neuronal number/density and proportion of cell types (progenitor vs neuron vs glial cells). The authors now provide some western blot data, but this not provide quantification of proportion of cell type, only protein expression levels. Moreover even in this case they do find differences for UK5099 and CB839 on levels of NeuN and GFAP. As requested before, the authors should still provide some data on proportion of cell types in these conditions using immunostaining for cell fate markers. If they do find differences in this case, this should be discussed explicitly in the manuscript.
3. The authors have solidified the data on dendritic and spine morphogenesis, but in Fig.8D, from the pictures shown the UK5099 seems to induce a massive change in the morphology of the spines (almost all of them appear to be thin and/or filopodia-like). This is not reflected by the quantifications in which these types of spines increase but remain a minority. Conversely mushroom spines are not visible on the picture of the UK5099 condition, while in the quantification this type of spines appears to constitute close to 50% of them. The authors should adapt the representative pictures or reassess their quantifications or.
4. The new experiments examining the effects of glutamate receptors agonists and antagonists on translation rates is a very welcome addition. However it remains that glutamate addition alone does not lead to translation rescue following UK5099 treatments, and the effects of glutamate receptor antagonists together with glutamate were not contributive. The authors should mention and discuss these partially contradictory data, thereby providing a more balanced and careful interpretation of their results.

Referee #3:

The authors have addressed almost all of my concerns.

1. The remaining concern that the authors should address is the statistics. While it may be that "...it is common practice to select several neurons from a few cultures to perform imaging analysis..." I insist that this common practice is not the best statistical practice because it inflates the number of data points. I therefore urge the authors to use the average from an individual experiment to do the statistics. If the effect is robust, taking the average will not change the outcome.
2. Moreover, I suggest that the authors change the title from "Glucose-derived glutamate drives neuronal terminal differentiation" to "Glucose-derived glutamate drives in vitro neuronal terminal differentiation" as they present largely in vitro evidence for the relevance of the pathway.

Referee #1:

The authors have performed additional experiments for the revision. I believe all of my questions have been addressed quite well.

We thank the reviewer for appreciating our revised manuscript.

Referee #2:

The authors have addressed many of the points raised in the first round of review, but some important points remain unaddressed, as explained below.

1. It is appreciated that it will be difficult for the authors to obtain in vivo evidence. However in this case the fact that all data obtained are in vitro data should be specified in the title or abstract, as requested before.

In the revised version of the manuscript, we changed the title as requested ("Glucose-derived glutamate drives neuronal terminal differentiation in vitro").

2. As requested before, given that metabolism can have pleiotropic effects during development, the authors should provide more data on the impact of the chemical compounds used on neuronal number/density and proportion of cell types (progenitor vs neuron vs glial cells). The authors now provide some western blot data, but this not provide quantification of proportion of cell type, only protein expression levels. Moreover even in this case they do find differences for UK5099 and CB839 on levels of NeuN and GFAP. As requested before, the authors should still provide some data on proportion of cell types in these conditions using immunostaining for cell fate markers. If they do find differences in this case, this should be discussed explicitly in the manuscript.

We thank the reviewer for his/her suggestion, and we agree that western blot analysis does not provide a clear information regarding the proportion of cell types. We performed additional experiments to address this issue. We first performed an immunostaining to detect the different cellular populations in our primary cultures at DIV1 and DIV14 using the following cell fate markers: GFAP for astrocytes, MAP2 for neurons, Nestin for progenitors (we tested two different antibodies: anti-chicken Nestin from Abcam ab134017 and anti-rabbit Nestin gently provided by Professor Anna Maria Cariboni) and DAPI for nuclei. As shown in the figure below, in our cultures we couldn't find any Nestin-positive cell already at DIV1, indicating that progenitors are not present in our *in vitro* model. Furthermore, DIV1 neurons are MAP2 positive, indicating that they are already committed to the neuronal fate.

To investigate the proportion of neuron and astrocytes upon the administration of the different compounds used in this study, we treated DIV1 neurons with Etomoxir, CB839 or UK5099 and labeled cells with the astrocyte marker GFAP, the neuronal marker MAP2 and DAPI for the nuclei. All treatments didn't change the proportion of the two cell types

compared with the control condition (Appendix Figure S1). Accordingly, the metabolic effects observed are not dependent on changes in the proportion of the two cell populations.

A

B

Progenitors are not present in primary cultures

A-B Representative confocal images of Map2, GFAP, Nestin, and DAPI staining in DIV1 and DIV14 primary cultures. Two different antibodies were used to stain progenitors (anti-chicken Nestin Abcam ab134017 in (A) and anti-rabbit Nestin, gently provided by Professor Anna Maria Cariboni (B)). Any Nestin positive cell was present in our culture (Map2: grey, GFAP: magenta, Nestin: yellow, DAPI: Cyan; scale bar 50 μ m).

UNIVERSITÀ DEGLI STUDI DI MILANO

DIPARTIMENTO DI SCIENZE FARMACOLOGICHE
E BIOMOLECOLARI "RODOLFO PAOLETTI" - DiSFeB

Direttore: Prof.ssa Monica DiLuca

3. The authors have solidified the data on dendritic and spine morphogenesis, but in Fig.8D, from the pictures shown the UK5099 seems to induce a massive change in the morphology of the spines (almost all of them appear to be thin and/or filopodia-like). This is not reflected by the quantifications in which these types of spines increase but remain a minority. Conversely mushroom spines are not visible on the picture of the UK5099 condition, while in the quantification this type of spines appears to constitute close to 50% of them. The authors should adapt the representative pictures or reassess their quantifications or.

We agree with the reviewer that the image of UK5099 dendritic spines in Figure 8D was not fully representative. In the revised version of the manuscript, we changed the picture to better show differences.

4. The new experiments examining the effects of glutamate receptors agonists and antagonists on translation rates is a very welcome addition. However it remains that glutamate addition alone does not lead to translation rescue following UK5099 treatments, and the effects of glutamate receptor antagonists together with glutamate were not contributive. The authors should mention and discuss these partially contradictory data, thereby providing a more balanced and careful interpretation of their results.

The glutamate supplementation induced a partial rescue on protein synthesis in dendritic spines as well as synaptoneurosomes of UK5099 neurons (Figure 7 D-E), without any effect on global translation. Furthermore, glutamate supplementation fully restored PSD-95 protein levels at the synapse, in line with our findings that prove a significant improvement of synaptic transmission upon glutamate supplementation (Figure 5D). This evidence indicated that glutamate contributes to the local control of the expression of synaptic proteins, without influencing the translation rate in cell body.

Glutamate can activate different receptors and trigger several signalling pathways that can differently affect protein translation rate. For instance, N-methyl-D-aspartic acid receptor (NMDAR) stimulation globally attenuates protein synthesis at synapse (Marin et al., J. Neuroscience 1997; Luchelli et al., J. Cell Science 2015; Sheetz et al. Nature Neuroscience 2000).

Therefore, we focused on glutamate metabotropic receptors (mGluR) because of their well-known role in promoting protein translation at the synapse. Moreover, we proved in our system that the antagonist of mGluR (MPEP) significantly reduced the protein translation and PSD-95 protein levels to an extent comparable to UK5099 (Figure EV5B).

We performed the experiments suggested by the reviewer using MPEP in cultures exposed to UK5099 and Glutamate. The combination of all these drugs had a dramatic effect on cell viability, as reported in the previous reply to the reviewer. UK5099 significantly affected protein translation rate and the glutamate supplementation partially rescued this effect. Therefore, blocking the mGluR through MPEP exposure prevented the

UNIVERSITÀ DEGLI STUDI DI MILANO

DIPARTIMENTO DI SCIENZE FARMACOLOGICHE
E BIOMOLECOLARI "RODOLFO PAOLETTI" - DiSFeB

Direttore: Prof.ssa Monica DiLuca

positive effect of glutamate supplementation on protein translation (in 3 out of 5 experiments, protein translation is almost abolished in UK5099+glutamate+MPEP cells), thus affecting cell viability.

In light of these results, we preferred to adopt another strategy to demonstrate the role of glutamate as neurotransmitter able to restore local protein translation in UK5099-treated cells. We provided a supplementation of DHPG, an activator of mGluR, that clearly showed a full rescue of protein local translation in UK5099-exposed cells (Figure 8B).

Therefore, we believe that our results support the role of glutamate in the rescue of protein translation following UK5099 treatment because:

- ✓ The glutamate supplementation induced a partial rescue on protein synthesis in dendritic spines as well as synaptoneurosomes of UK5099 neurons (Figure 7 D-E).
- ✓ The inhibition of mGluR phenocopied the effects of UK5099 on global protein translation and on PSD-95 proteins levels (Figure EV5B).
- ✓ Blocking the mGluR through MPEP exposure prevented the positive effect of glutamate supplementation on protein translation and affected cell viability.
- ✓ DHPG, an activator of mGluR fully rescued local protein translation in UK5099-exposed cells (Figure 8B).

We have provided specific comments in the revised manuscript to clarify these issues.

Effect of MPEP administration on global protein synthesis in neurons exposed to UK5099 supplemented with glutamate.

Representative western blot and quantification of puromycin incorporation in total extract of DIV15 rat primary neurons treated at DIV1 with UK5099 and UK5099 supplemented with Glutamate in presence/absence of MPEP. β -Actin was used as loading control. Data

UNIVERSITÀ DEGLI STUDI DI MILANO

DIPARTIMENTO DI SCIENZE FARMACOLOGICHE
E BIOMOLECOLARI "RODOLFO PAOLETTI" - DiSFeB

Direttore: Prof.ssa Monica DiLuca

are shown as mean \pm SE (one way ANOVA, Tukey's multiple comparison test. n= 5 independent experiments DMSO vs UK5099 p=0,0188, DMSO vs UK5099+Glu p=0,2100, DMSO vs UK5099+Glu + MPEP p=0,0386; UK5099 vs UK5099 +Glu p=0,9885; UK5099 vs UK5099+ Glu + MPEP p=0,5982, UK5099+ Glu vs UK5099 + glu +MPEP p=0,0175

Referee #3:

The authors have addressed almost all of my concerns.

1. The remaining concern that the authors should address is the statistics. While it may be that "...it is common practice to select several neurons from a few cultures to perform imaging analysis...." I insist that this common practice is not the best statistical practice because it inflates the number of data points. I therefore urge the authors to use the average from an individual experiment to do the statistics. If the effect is robust, taking the average will not change the outcome.

We carried out the statistical analysis accordingly to the reviewer request and changed the graphs in the revised version of the manuscript.

2. Moreover, I suggest that the authors change the title from "Glucose-derived glutamate drives neuronal terminal differentiation" to "Glucose-derived glutamate drives in vitro neuronal terminal differentiation" as they present largely in vitro evidence for the relevance of the pathway.

We changed the title according to the reviewer suggestion.

Prof. Elena Marcello
University of Milano
Pharmacological and Biomolecular Sciences
Via Balzaretti 9
Milano, N/A 20133
Italy

Dear Prof. Marcello,

I am very pleased to accept your manuscript for publication in the next available issue of EMBO reports. Thank you for your contribution to our journal.

Yours sincerely,
